# Inducing cancer indolence by targeting mitochondrial Complex I is potentiated by blocking macrophage-mediated adaptive responses

Ivana Kurelac[1,2], Luisa Iommarini [3], Renaud Vatrinet[1,3], Laura Benedetta Amato[1], Monica De Luise[1], Giulia Leone[3], Giulia Girolimetti[1], Nikkitha Umesh Ganesh[1], Victoria Louise Bridgeman[2], Luigi Ombrato[2], Marta Columbaro[4], Moira Ragazzi[5], Lara Gibellini[6], Manuela Sollazzo[3], Rene Gunther Feichtinger[7], Silvia Vidali [7], Maurizio Baldassarre[1], Sarah Foriel[8,9], Michele Vidone[1], Andrea Cossarizza[6], Daniela Grifoni [3], Barbara Kofler[7], Ilaria Malanchi[2], Anna Maria Porcelli[3,10] & Giuseppe Gasparre[1,11]

Converting carcinomas in benign oncocytomas has been suggested as a potential anti-cancer strategy. One of the oncocytoma hallmarks is the lack of respiratory complex I (CI). Here we use genetic ablation of this enzyme to induce indolence in two cancer types, and show this is reversed by allowing the stabilization of Hypoxia Inducible Factor-1 alpha (HIF-1α). We further show that on the long run CI-deficient tumors re-adapt to their inability to respond to hypoxia, concordantly with the persistence of human oncocytomas. We demonstrate that CI-deficient tumors survive and carry out angiogenesis, despite their inability to stabilize HIF-1α. Such adaptive response is mediated by tumor associated macrophages, whose blockage improves the effect of CI ablation. Additionally, the simultaneous pharmacological inhibition of CI function through metformin and macrophage infiltration through PLX-3397 impairs tumor growth in vivo in a synergistic manner, setting the basis for an efficient combinatorial adjuvant therapy in clinical trials.

[1] Dipartimento di Scienze Mediche e Chirurgiche, Università di Bologna, Via Massarenti 9, 40138 Bologna, Italy. [2] Tumor-Host Interaction Lab, The Francis Crick Institute, 1 Midland Rd, NW1 1AT London, UK. [3] Dipartimento di Farmacia e Biotecnologie, Università di Bologna, Via Selmi 3, 40126 Bologna, Italy. [4] Laboratory of Musculoskeletal Cell Biology, IRCCS Istituto Ortopedico Rizzoli, Via Giulio Cesare Pupilli 1, 40136 Bologna, Italy. [5] Anatomia Patologica, Azienda Ospedaliera S. Maria Nuova di Reggio Emilia, Viale Risorgimento 80, 42123 Reggio Emilia, Italy. [6] Dipartimento di Scienze Mediche e Chirurgiche materno infantili e dell'adulto, Università degli Studi di Modena e Reggio Emilia, Via del Pozzo 71, 41124 Modena, Italy. [7] Research Program for Receptor Biochemistry and Tumor Metabolism, Department of Pediatrics, University Hospital of the Paracelsus Medical University, Muellner Hauptstraße 48, 5020 Salzburg, Austria. [8] Khondrion BV, Philips van Leydenlaan 15, 6525 EX Nijmegen, The Netherlands. [9] Radboud Center for Mitochondrial Medicine (RCMM) at the Department of Pediatrics, Radboud University Medical Center, Geert Grooteplein Zuid 10, 6500 HB Nijmegen, The Netherlands. [10] Centro Interdipartimentale di Ricerca Industriale Scienze della Vita e Tecnologie per la Salute, Università di Bologna, Via Tolara di Sopra 41/E, 40064 Ozzano dell'Emilia, Italy. [11] Centro di Ricerca Biomedica Applicata (CRBA), Università di Bologna, Via Massarenti 9, 40138 Bologna, Italy. These authors jointly supervised this work: Anna Maria Porcelli, Giuseppe Gasparre. Correspondence and requests for materials should be addressed to I.M. (email: ilaria.malanchi@crick.ac.uk) or to A.M.P. (email: annamaria.porcelli@unibo.it) or to G.G. (email: giuseppe.gasparre3@unibo.it)

Developing therapeutic strategies to target cancer metabolism is currently gaining momentum and one of the rising star metabolic approaches displaying antineoplastic potential involves inhibition of respiratory complex I (CI)[1–3], the first and rate-limiting enzyme of oxidative phosphorylation (OXPHOS). A profound revisiting of the seminal Warburg's hypothesis that tumors rely on aerobic glycolysis to fuel growth has led to establish a fundamental role for mitochondrial respiration in cancer progression. It is now accepted that highly aggressive, malignant cancer cells combine glycolytic and mitochondrial metabolic routes to meet energetic and biosynthetic demands[4]. Indeed, to maintain mitochondrial respiration, aggressive human cancers usually counterselect pathogenic mitochondrial DNA (mtDNA) CI mutations[5–8]. Conversely, severe mtDNA CI mutations are found in indolent, low-proliferative oncocytic tumors[8,9], i.e., neoplasms characterized by cells accumulating mostly dysfunctional, aberrant mitochondria and displaying scarce vasculature associated with destabilization of Hypoxia Inducible Factor-1 alpha (HIF-1α), the main promoter of vasculogenesis, glycolysis, and survival in hypoxic environment[10]. Oncocytomas represent an excellent case study in oncology, as they appear to be de facto short-circuited tumors that have become confined to a low-proliferative state due to metabolic constraints, likely deriving from the occurrence of high loads of pathogenic mtDNA mutations[8,11] or from an impairment in autophagy[12]. Converting carcinomas into oncocytomas as an anti-cancer strategy has been proposed by targeting autophagy master regulator ATG7[13]. Starting from the identification of genetic hallmarks of oncocytomas, i.e., severe mtDNA mutations in CI, targeting this enzyme may be an even more efficient alternative approach to induce indolence, as this would simultaneously cause OXPHOS defects and the inability to adapt to hypoxia, shutting off several essential pathways in cancer cells. However, even if the severe CI damage could be expected to cause a metabolic catastrophe and impede malignant progression, oncocytic tumors linger in their indolent and slow-growing state, displaying quiescent but potentially perilous features of chemoresistance[14,15]. This is evident in human neoplasms rather than in mouse models, where reversion of the benign phenotype is technically difficult to assess for such slow-growing cancers. It is therefore yet unclear how human CI-deficient tumors may promote angiogenesis despite HIF1 impairment. Thus, since modes of re-adaptation to CI dysfunction seem to exist, the identification of key factors keeping cancer cells alive is mandatory to design efficient combinatorial strategies to eradicate tumors. At the same time, to provide full justification for the use of CI inhibitors such as metformin in clinical practice, the dissection of the mechanisms linking CI inhibition to cancer growth arrest is warranted, especially those behind HIF-1α destabilization.

To fill the aforementioned gaps, we generated cancer cell lines lacking CI, via knockout of nuclear-encoded CI core subunit NDUFS3. Disengaging from the technical difficulties of dealing with mtDNA genetics, these models allow fine-tuning of NDUFS3 levels and subsequent CI activity.

We provide the proof of concept that CI ablation reduces tumorigenic potential and allows conversion into low-proliferative oncocytoma. Furthermore, while proving that the loss of HIF-1α is accountable for the decreased tumorigenic potential upon targeting CI, we additionally discover an atypical microenvironment response mediated by protumorigenic macrophages, which support survival of CI-deficient masses, and which we synergistically targeted to significantly increase therapeutic efficacy of metformin.

## Results

**NDUFS3 knockout induces a low-proliferative cancer phenotype.** The molecular mechanisms linking CI impairment and reduction of tumorigenic potential have only been partially addressed, mainly due to difficult-to-handle cell models bearing mtDNA mutations. Thus, with the aim to demonstrate the anti-tumorigenic effect of CI deficiency and investigate the underlying mechanisms, we first generated easier-to-handle cancer cell models bearing a mtDNA-independent CI dysfunction. The nuclear-encoded NDUFS3 gene was knocked-out to induce CI deficiency in mesenchymal (osteosarcoma 143B) and epithelial (colorectal cancer HCT116) cancer cells, hereafter referred to as 143B$^{-/-}$ and HCT$^{-/-}$ (Supplementary Fig. 1), with the aim to generalize our findings in two cancer models of different tissue origin. Genetic ablation of NDUFS3 induced a severe decrease of CI NADH dehydrogenase activity (Supplementary Fig. 1d), reduced CI-driven ATP production (Supplementary Fig. 2a) and blocked mitochondrial respiration (Supplementary Fig. 2b), resulting in a major OXPHOS defect. On the other hand, NDUFS3 knockout induced an increase in glucose consumption (Supplementary Fig. 2c) and lactate production (Supplementary Fig. 2d), indicating that upregulation of glycolysis compensates the CI defect in vitro. Concordantly, the total amount of ATP was found to be comparable in CI-competent and deficient cells (Supplementary Fig. 2e). Moreover, the α-ketoglutarate (α-KG)/citrate ratio, as well as the m + 5 citrate enrichment after labeling with $^{13}$C-labeled glutamine, were markedly increased in CI-deficient cells, irrespective of the cellular background (Supplementary Fig. 2f–g). These data indicate that CI ablation triggers the preferential use of reductive carboxylation to supply the citrate pool, in agreement with previous studies in cancer cells with severe mtDNA mutations or treated with metformin[16–18]. Under selective pressures in vivo, the lack of NDUFS3 caused a significant decrease of xenograft growth in both cancer types (Fig. 1a) and reduced cell invasion (Fig. 1b), indicating that the metabolic reprogramming occurring following CI disruption (Supplementary Fig. 3a) allows cell survival but is not sufficient to fully recover the tumorigenic and invasive potential. 143B$^{-/-}$ and HCT$^{-/-}$ tumors displayed lower KI-67 proliferation index (Fig. 1c) and no signs of necrosis, a known marker of aggressiveness, which was instead abundant in CI-competent tumors (Fig. 1d). The cytostatic effect of CI ablation was confirmed by a decreased caspase activation in the KO tumors, in agreement with what we previously reported[19] (Supplementary Fig. 3b). Interestingly, CI-deficient masses were composed of eosinophilic cancer cells (Fig. 1d) harboring swollen mitochondria with deranged cristae (Fig. 1e, Supplementary Fig. 3c), well recapitulating oncocytic lesions, which was further strengthened by the finding that 143B$^{-/-}$ xenografts displayed an increased expression of PPARGC1A, the master regulator of mitochondrial biogenesis (Supplementary Fig. 3d). Re-expression of NDUFS3 in 143B$^{-/-}$ cells rescued CI activity (Supplementary Fig. 3e), mitochondrial ultrastructure and tumorigenic potential (Fig. 1f). Taken together, these data indicate that NDUFS3 knockout tumors recapitulated the bioenergetic features of lesions harboring mtDNA disruptive mutations, such as oncocytes, and provide the proof of principle for the anti-tumorigenic effect of CI damage.

**Targeting CI converts malignant cancers in oncocytomas.** After establishing that tumorigenic potential of CI-deficient tumors is reduced compared to their CI-competent counterpart, we sought to prove that depleting CI in a well-formed, progressing mass may represent an effective strategy to decrease or arrest growth in vivo. NDUFS3 was constitutively expressed in 143B$^{-/-}$ cells

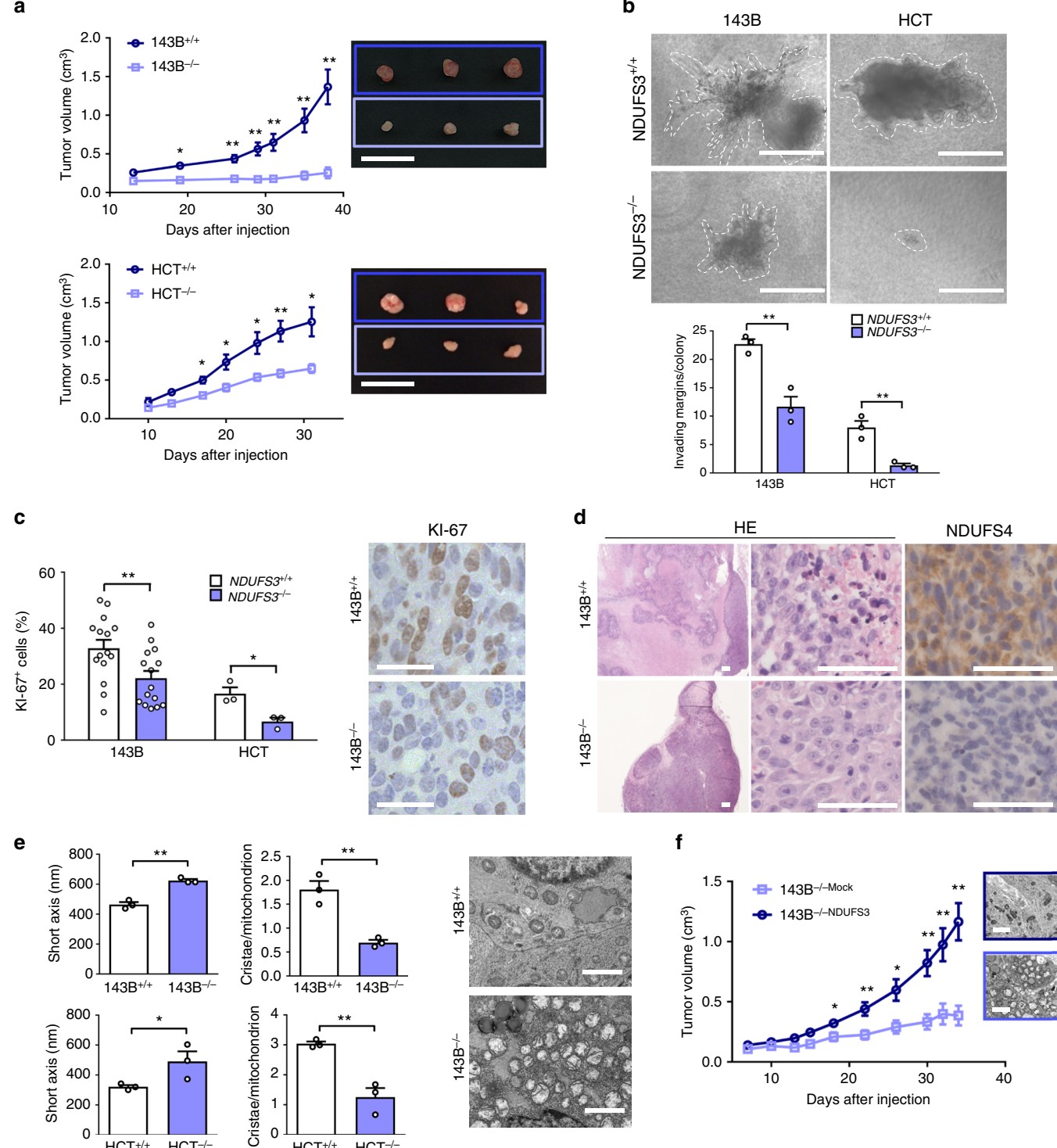

(143B$^{-/-}$NDUFS3), under the control of a transactivator negatively regulated by doxycycline (Dox) (Supplementary Fig. 4a). By using Dox doses which did not affect CI assembly, a complete loss of the enzyme was observed after 9 days of treatment (Supplementary Fig. 4b). Moreover, at doses which do not impact tumor growth or mtDNA-encoded protein levels (Supplementary Fig. 4c–e), Dox-mediated induction of CI loss caused growth arrest in 85.7% of 143B$^{-/-}$NDUFS3 proliferating tumors, decreased KI-67 proliferative index and resulted in significantly prolonged survival of the animals (Fig. 2a, b, Supplementary Fig. 5). CI-depleted tumors showed clear-cut negative

NDUFS3 staining (Fig. 2c), demonstrating complete loss of the protein and suggesting that no promotor leakage occurred. Staining for NDUFS4 subunit, a marker of CI assembly[9], was reduced in Dox-treated 143B$^{-/-}$NDUFS3 xenografts, indicating that the enzyme was successfully targeted (Fig. 2c). Furthermore, Dox-treated masses displayed ultrastructural mitochondrial alterations (Fig. 2d) and lack of necrosis at tumor leading margins (Fig. 2e), a phenotype resembling *NDUFS3* knockout xenografts. All these data demonstrate that targeting CI during cancer progression converts malignant tumors into low-proliferative, oncocytoma-like tumors.

**Fig. 1** *NDUFS3* knockout triggers oncocytic phenotype and reduces tumorigenic potential. **a** Growth curves of 143B xenografts in CD-1 nude mice [$n = 15$, df $= 28$, $t$(day 38) $= 5.4$] and HCT xenografts in ICRF nude mice [$n = 8$, df $= 14$, $t$(day 31) $= 3.6$]. Data are mean ± s.e.m. Representative tumors are shown. Scale bars: 2 cm. **b** Cell invasion assay in 3D collagen-I/Matrigel media of 143B and HCT xenograft-derived cells. Representative colonies are displayed and their margin delineated with dashed line. Scale bars: 400 µm. Quantification of invading areas of the colonies is shown [$n = 3$, df $= 4$, $t$(143B) $= 5.578$, $t$(HCT) $= 5.547$]. Data are mean ± s.e.m. **c** Quantification of cells displaying KI-67 positive nuclei in 143B ($n = 15$, df $= 28$, $t = 2.7$) and HCT ($n = 3$, df $= 4$, $t = 3.7$) tumors. Representative image of KI-67 immunohistochemistry staining is shown. Scale bars: 25 µm. Data are mean + s.e.m. **d** Hematoxylin/eosin (HE) staining and immunohistochemistry for NDUFS3 and NDUFS4 CI subunits. Representative images are shown. Scale bars: 50 µm. **e** Evaluation of short mitochondrial axis width and *cristae* number in 143B xenografts (axis: $n = 3$, df $= 4$, $t = 8.3$; cristae: $n = 3$, df $= 4$, $t = 5.9$) and HCT cells (axis: data were log-transformed, $n = 3$, df $= 4$, $t = 2.8$; cristae: $n = 3$, df $= 4$, $t = 5.7$). Representative electron micrographs are shown. Scale bars: 2 µm. Data are mean + s.e.m. **f** Growth curves of 143B xenografts derived from cells carrying the empty vector (143B$^{-/-Mock}$) or re-expressing wild-type NDUFS3 (143B$^{-/-NDUFS3}$) in CD-1 nude mice. Data are mean ± s.e.m. [$n = 6$, df $= 10$, $t$(day 34) $= 5.2$]. Representative electron micrographs of the xenografts are shown. Scale bars: 2 µm. In each panel, statistical significance is specified with asterisks (*$p < 0.05$, **$p < 0.01$, ***$p < 0.001$)

Next, we corroborated our findings in a *Drosophila melanogaster* model of tumorigenesis, in which CI abolishment was feasible through the targeting of another crucial CI subunit, namely the ortholog of human *NDUFV1*. This allowed us to investigate whether the phenotype observed by targeting *NDUFS3* might be extrapolated to any severe CI damage which would hamper specific NADH dehydrogenase activity, leading to NADH accumulation, as we previously showed[19–21]. A polarity-deficient strain was used, in which mutations of the evolutionarily conserved *lethal giant larvae* (*lgl*) tumor suppressor gene induce malignant growth of the epithelial larval organs which progressively grow into frank cancers, causing late individuals to die as pre-pupae with their anterior half completely filled by giant tumors[22]. In this transformed context, we induced *GFP⁺lgl⁻/⁻ndufv1* knockdown clones (hereafter referred to as *V1*$^{KD}$) at 6 days after egg laying (AEL), a time when neoplastic growth is obvious. As a control, *GFP⁺lgl⁻/⁻luc*$^{KD}$ clones were induced in the same experimental conditions. The epithelial tumors were isolated from GFP⁺ larvae at 8 days AEL. The CI-deficient *V1*$^{KD}$ masses were smaller than controls (Fig. 2f, Supplementary Fig. 6a) and cell density analysis revealed an average of 50.75% GFP⁺ cells in the *V1*$^{KD}$ samples versus 74.87% in the control counterparts (Fig. 2f). Subsequently, to investigate the phenotypic consequences of CI disruption in cells undergoing neoplastic transformation in an otherwise wild-type organ, mimicking mammalian cancer onset, we induced *ndufv1* knockdown in a cooperative system, extensively used to recapitulate clonal carcinogenesis in *Drosophila*, where a *lgl* loss of function mutation is combined with oncogenic *Ras* (*Ras*$^{V12}$)[23]. The *lgl⁻/⁻Ras*$^{V12}$ clones in which *ndufv1* was knocked-down appeared considerably smaller than the control counterparts and showed a lower roundness coefficient (Supplementary Fig. 6b–c), indicating poor capacity to form three-dimensional structures. The mitotic index measured by PH3 staining was reduced in *lgl⁻/⁻Ras*$^{V12}$*V1*$^{KD}$ tumors (Supplementary Fig. 6d), whereas cell death, defined by activation of caspase 3, was comparable between the *lgl⁻/⁻Ras*$^{V12}$*V1*$^{KD}$ and control samples (Supplementary Fig. 6e). These data confirmed that CI deficiency reduces tumorigenic potential by decreasing the proliferation rate of transformed cells, rather than inducing apoptosis, in line with our previous observations[19]. Altogether, the data collected in the two *Drosophila* cancer models sustained the findings obtained in the mammalian systems, confirming the essential role of CI in both initiation and progression of tumor growth, and indicating CI as a valid target for anti-cancer therapy.

**HIF-1α loss is involved in CI-null tumor growth decrease.** CI deficit in oncocytic tumors has been associated with lack of HIF-1α stabilization[10], a phenomenon also observed in xenografts carrying severe mtDNA mutations[19,21] or treated with metformin[1]. Indeed, HIF-1α stabilization was absent in the 143B⁻/⁻ and HCT⁻/⁻ tumors (Fig. 3a) and associated with a lower expression of HIF1-responsive genes (Fig. 3b). Moreover, 143B$^{-/-NDUFS3}$ xenografts, in which CI function was restored, showed the recovery of HIF-1α activation, whereas Dox-induced CI loss caused HIF-1α destabilization (Supplementary Fig. 7a). Finally, staining the *Drosophila* HIF-1α ortholog *Similar* showed lack of its nuclear accumulation in *lgl⁻/⁻Ras*$^{V12}$*V1*$^{KD}$, which in the control clones was evident and followed by expression of invasion marker MMP1 (Fig. 3c), confirming HIF-1α destabilization as an evolutionarily conserved mechanism associated with CI deficiency. Interestingly, CI-deficient xenografts were prevalently negative for pimonidazole staining, used as the hypoxia marker, even when left to grow until the size comparable to that of CI-competent controls (10% of the animal weight) (Supplementary Fig. 7b, c), suggesting that CI-deficient tumors barely ever acquire a hypoxic phenotype. The rare hypoxic foci identified in 143B⁻/⁻ xenografts were negative for HIF-1α staining (Fig. 3d), allowing to hypothesize that CI deficiency causes HIF-1α destabilization even in true hypoxia. To establish whether CI deficiency induces defects in HIF-1α stabilization, we analyzed HIF-1α status under controlled hypoxic environment in vitro. 143B⁻/⁻ and HCT⁻/⁻ cells grown under 1% O₂ conditions displayed a delayed HIF-1α stabilization (Fig. 3e) and lower expression of HIF1-responsive genes (Supplementary Fig. 7d), compared to their CI-competent counterparts, demonstrating an intrinsic defect in the HIF-1α stabilization pathway in CI-deficient tumors. With the aim to prove that the growth deficit of cells lacking functional CI was due to their inability to activate HIF1, a non-degradable and constitutively expressed form of HIF-1α[24], here named triple mutant (TM), was introduced into 143B⁻/⁻ and HCT⁻/⁻ cells. When compared to the empty vector controls (143B$^{-/-Mock}$ and HCT$^{-/-Mock}$), cells constitutively expressing the TM-HIF-1α (143B$^{-/-TM}$ and HCT$^{-/-TM}$) displayed HIF-1α stabilization, translocation into the nucleus and higher transcription of HIF1 downstream targets regardless of normoxia (Fig. 3f, Supplementary Fig. 8a, b), and formed larger colonies in 3D collagen/matrigel media (Supplementary Fig. 8c). Moreover, while TM-HIF-1α did not influence the tumorigenic potential of 143B⁺/⁺ cells (Supplementary Fig. 8d), CI-deficient cells expressing TM-HIF-1α formed larger xenografts when injected in nude mice (Fig. 3g, Supplementary Fig. 8e). Thus, the growth deficit of cells lacking functional CI was at least partially due to their inability to activate HIF1.

**CI deficiency boosts PHD activity and destabilizes HIF-1α.** We proceeded to explore the molecular mechanisms linking CI impairment with HIF-1α destabilization. The best-characterized mode of HIF-1α regulation is the PHDs-mediated hydroxylation, which in normoxia triggers its proteasomal degradation[25]. In addition, several PHD-independent mechanisms of HIF-1α regulation have been identified[26], such as reactive oxygen species

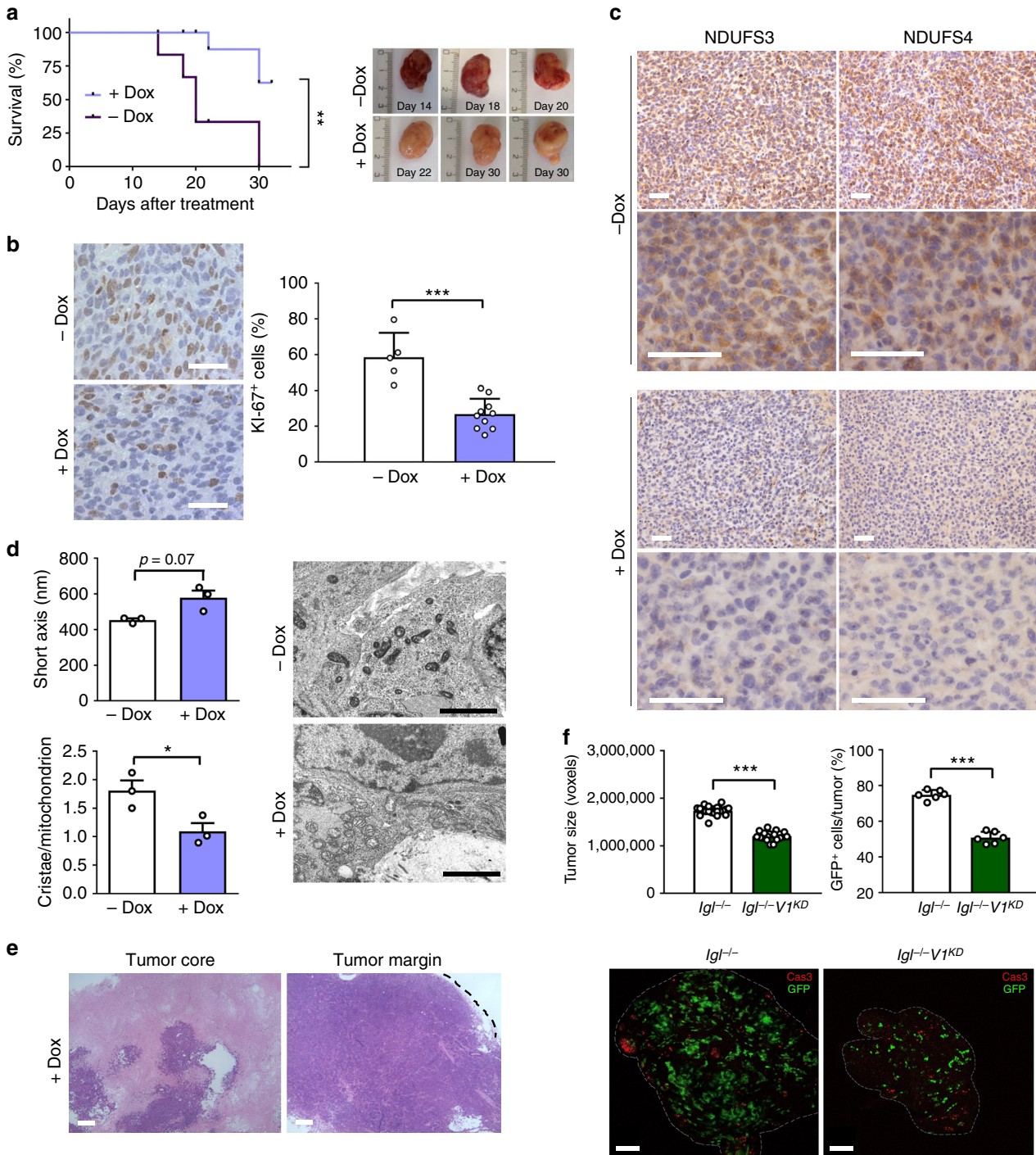

**Fig. 2** Targeting CI arrests tumor progression and converts carcinomas into oncocytomas. **a** Kaplan–Meier survival curves of CD-1 nude mice injected with 143B$^{-/-NDUFS3}$ cells and treated with ($n = 7$) or without ($n = 5$) Dox (1 mg mL$^{-1}$ in drinking water). Survival end-point: xenografts reaching 10% of animal weight. Representative xenografts are shown. **b** Quantification of cells displaying KI-67 positive nuclei in 143B$^{-/-NDUFS3}$ tumors treated with ($n = 7$) or without ($n = 5$) Dox (1 mg mL$^{-1}$ in drinking water) (df = 13, $t = 5.539$). Representative image of KI-67 immunohistochemistry staining is shown. Scale bars: 50 μm. Data are mean + s.e.m. **c** Immunohistochemistry staining of CI subunits in xenografts treated with or without Dox (1 mg mL$^{-1}$ in drinking water). Representative images are shown. Scale bars: 50 μm. **d** Short mitochondrial axis width ($n = 3$, df = 4, $t = 2.4$) and *cristae* number ($n = 3$, df = 4, $t = 3.1$) evaluation, with representative electron micrographs of xenografts treated with or without Dox (1 mg mL$^{-1}$ in drinking water). Scale bars: 2 μm. Data are mean + s.e.m. **e** Hematoxylin/eosin staining of 143B$^{-/-NDUFS3}$ tumors treated with Dox. Tumor margin is indicated by the dashed line. Representative images are shown. Scale bars: 100 μm. **f** Immunofluorescence for caspase 3 (Cas3) of CI-competent ($lgl^{-/-}$) and deficient ($lgl^{-/-}V1^{KD}$) epithelial tumors (GFP$^+$) of the fly. Dashed lines delineate tumor margin. Scale bars: 80 μm. Tumor size ($n = 25$, df = 48, $t = 13.2$) and the average number of GFP$^+$ cells ($n = 6$, df = 10, $t = 13.8$) were evaluated. Data are mean + s.e.m. In each panel, statistical significance is specified with asterisks (*$p < 0.05$, **$p < 0.01$, ***$p < 0.001$)

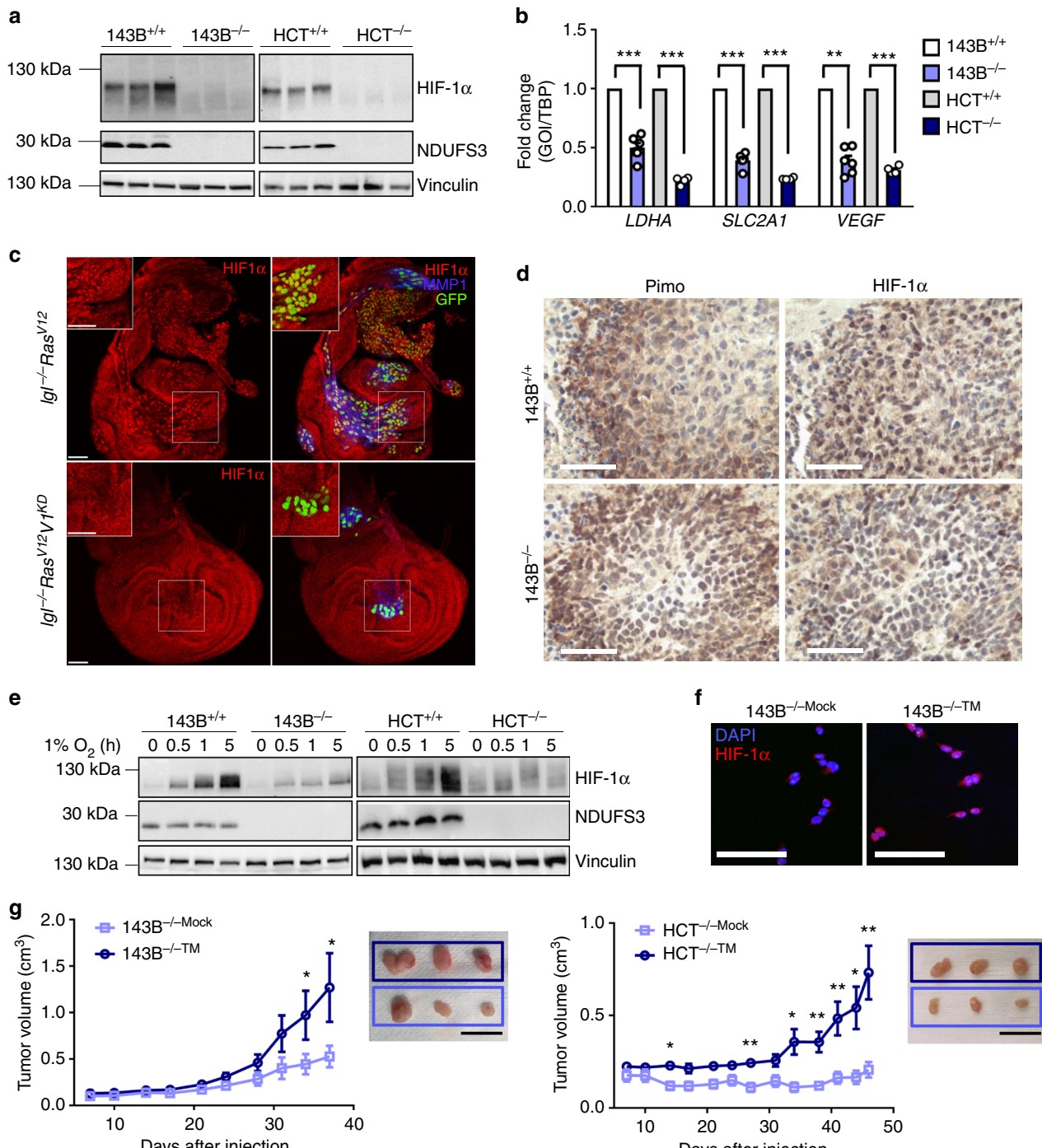

**Fig. 3** CI deficiency promotes HIF-1α destabilization. **a** HIF-1α and NDUFS3 western blot analysis in 143B and HCT xenografts. Vinculin was used as loading control. **b** Gene expression of HIF-1α targets evaluated by qRT-PCR in 143B [$n = 8$, df = 14, $t(LDHA) = 6.5$, $t(SLC2A1) = 4.5$, $t(VEGF) = 4.7$] and HCT [$n = 4$, df($LDHA$) = 6, $t(LDHA) = 14.1$, df($SLC2A1$) = 3, $t(SLC2A1) = 9.3$, df($VEGF$) = 6, $t(VEGF) = 11.7$] xenografts. For $SLC2A1$ in HCT cells the Student's $t$-test assuming unequal variance was applied. Data are mean + s.e.m. **c** Immunofluorescence for HIF-1α and MMP1 in *Drosophila* wing discs. Tumor tissue is composed of GFP+ cells. Representative images are shown. Scale bars: 40 μm. **d** Immunohistochemical staining of pimonidazole (Pimo)-labeled protein adducts and HIF-1α in 143B[+/+] and 143B[−/−] tumors of equivalent volume (2 cm³). Representative images are shown. Scale bars: 50 μm. **e** HIF-1α and NDUFS3 western blot of 143B and HCT cells cultured in 1% O₂ at indicated times. Vinculin was used as loading control. **f** Immunofluorescent staining of nuclei (DAPI) and HIF-1α in 143B[−/−] cells carrying empty vector (Mock) or TM-HIF-1α (TM). Representative images are shown. Scale bars: 100 μm. **g** Growth curves in CD-1 nude mice of 143B [$n = 15$, df = 28, $t(\text{day } 37) = 2.2$] and HCT [data were log-transformed, $n = 7$, df = 12, $t(\text{day } 46) = 3.5$] xenografts derived from CI-deficient cells carrying empty vector (Mock) or TM-HIF-1α (TM). Data are mean ± s.e.m. Representative tumors are shown. Scale bars: 2 cm. In each panel, statistical significance is specified with asterisks (*$p < 0.05$, **$p < 0.01$, ***$p < 0.001$)

(ROS)-mediated activation of PI3K/Akt or ERK pathways which promote *HIF1A* transcription and translation[27]. No significant differences in $H_2O_2$ production and, concordantly, in the total glutathione level, between CI-competent and deficient cells were observed (Supplementary Fig. 9a, b), and the latter displayed no decrease but rather an increase in *HIF1A* mRNA levels compared to their control counterparts (Fig. 4a), indicating that HIF-1α destabilization in CI-deficient cells does not involve mechanisms affecting *HIF1A* transcription. Furthermore, HIF-1α recovery was observed after treatment with the proteasome inhibitor MG132 (Fig. 4b), indicating that the lack of CI does not affect HIF-1α translation. In particular, MG132 caused higher accumulation of hydroxylated HIF-1α (HIF-1α-OH) in 143B−/− and HCT−/−

cells (Fig. 4b, c), indicating an enhanced PHD activity in CI-deficient models. Recovery of HIF-1α protein was also observed in 143B−/− and HCT−/− cells after treatment with dimethyloxallyl glycine (DMOG), a selective PHDs inhibitor[28] (Fig. 4b), confirming CI deficiency determines a deregulation in the PHD-mediated degradation of the protein. Of note, PHD1/2 protein levels were comparable between CI-competent and deficient cells (Supplementary Fig. 9c). In agreement with our previous data showing NADH-mediated α-KG accumulation in tumors carrying CI-disruptive mtDNA mutations[19,21], CI-deficient cells displayed a significant increase in α-KG levels compared to controls, as well as in malate/aspartate ratio (Fig. 4d), the latter being an indication of NADH accumulation. α-KG is known to boost PHD

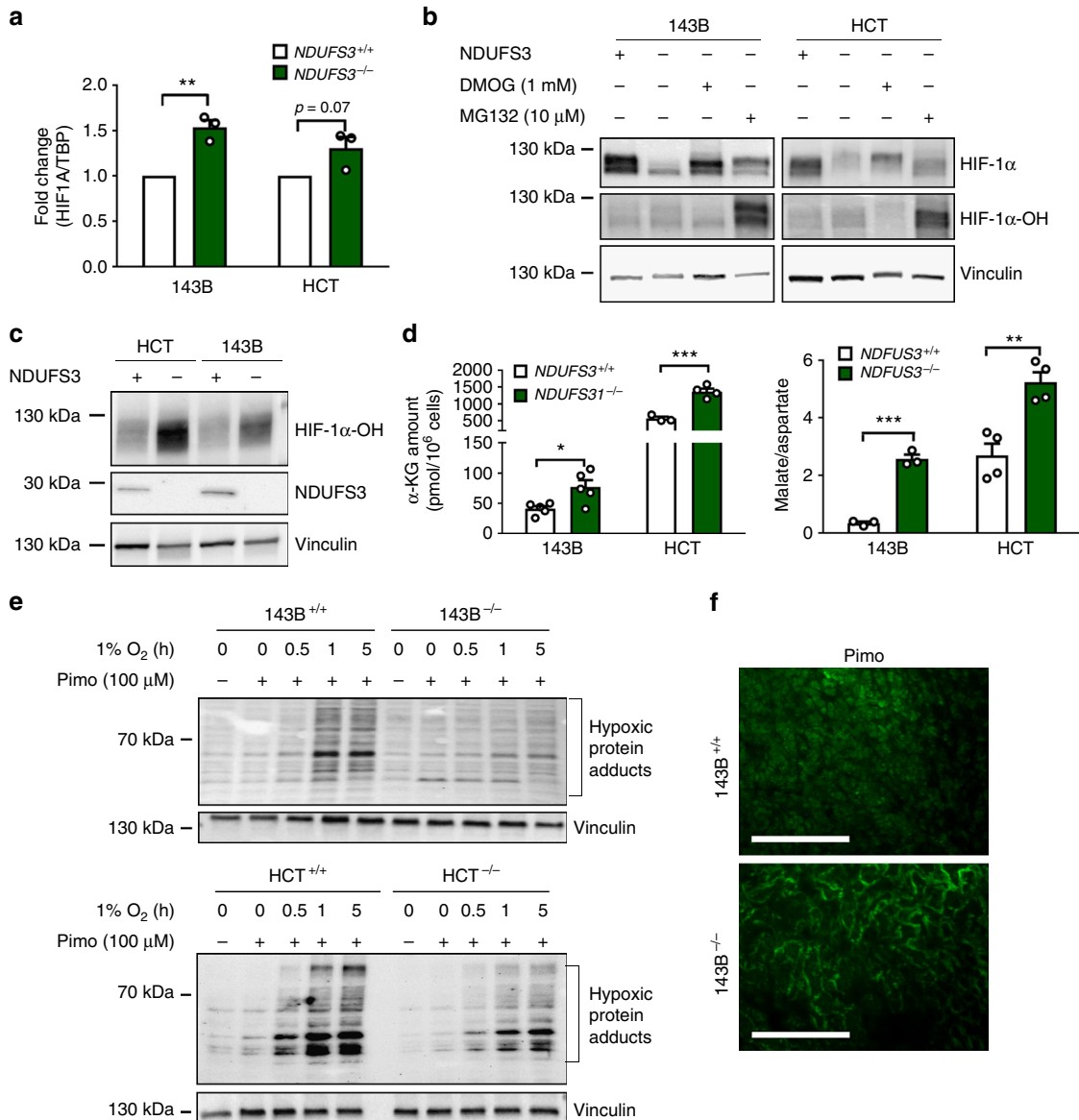

**Fig. 4** CI deficiency induces HIF-1α destabilization via increased PHD activity. **a** *HIF1A* expression level evaluated by qRT-PCR in 143B ($n = 3$, df = 4, $t = 5.2$) and HCT ($n = 3$, df = 4, $t = 3.1$) cells cultured for 5 h in 1% $O_2$. Data are mean + s.e.m. **b** HIF-1α and HIF-1α-OH western blot analysis in 143B and HCT cells exposed to hypoxia (1% $O_2$ for 5 h) and treated with DMOG (1 mM for 3 h) or MG132 (10 μM for 3 h). Vinculin was used as loading control. **c** HIF-1α-OH and NDUFS3 western blot analysis in 143B and HCT cells exposed to hypoxia (1% $O_2$ for 5 h) and treated with MG132 (10 μM for 3 h). Vinculin was used as loading control. **d** α-KG amount and malate/aspartate ratio in 143B [$n$(α-KG) = 2; $n$(malate/aspartate) = 3, df(malate/aspartate) = 4, $t$(malate/aspartate) = 14.3] and HCT [$n$ = 4, df = 6, $t$(α-KG) = 6.8, $t$(malate/aspartate) = 4.8] cells measured under basal conditions. Data are mean + s.e.m. **e** Protein adducts western blot analysis in cells incubated with pimonidazole (Pimo) (100 μM for 1 h). Vinculin was used as loading control. **f** Representative images of pimonidazole (Pimo) immunofluorescent staining in 143B xenografts. Scale bars: 100 μm. In each panel, statistical significance is specified with asterisks (**$p < 0.01$, ***$p < 0.001$)

activity by increasing its affinity for oxygen[29,30]. Furthermore, it has been shown that severe mitochondrial damage may cause a sufficient reduction in oxygen consumption to increase cytosolic oxygen concentrations regardless of the external hypoxia[31]. Thus, we evaluated the intracellular oxygen status of CI-deficient cells with the hypoxia marker pimonidazole and observed that, when grown in controlled in vitro hypoxic conditions, $143B^{-/-}$ and $HCT^{-/-}$ cells experienced less intrinsic hypoxia than their CI-competent counterparts (Fig. 4e), in accordance with the mainly normoxic condition observed in CI-deficient xenografts (Fig. 4f and Supplementary Fig. 7c). Taken together, these data imply that the delay and reduction of HIF-1α stabilization in CI-deficient cancer cells was caused by simultaneous increase in α-KG concentrations and reduced oxygen consumption, which together lead to a boost of PHD activity regardless of external hypoxia.

**CI-deficient tumors display stroma-associated angiogenesis.** Despite the lack of HIF-1α and lower tumorigenic potential, CI-deficient masses, like oncocytomas, continued to progress (Supplementary Fig. 7b), meaning they probably engage alternative, HIF-1α-independent mechanisms to ensure nutrient supply. In the context of proposing CI targeting as an anti-cancer strategy, such growth persistence might present a potential risk of malignancy. Thus, we next investigated the consequences of HIF-1α destabilization on angiogenesis in CI-deficient tumors, with the aim to identify the pro-survival adaptive mechanisms activated during their development. By staining with endothelial markers endomucin and CD31, we compared the vessels morphology in CI-competent and deficient masses. CI-competent tumors presented with vessels characterized by a large lumen and positivity for the pericyte marker smooth muscle actin (SMA), indicating mature and well-perfused vasculature (Fig. 5a). Moreover, these vessels were located in areas surrounded by hypoxia and HIF-1α-positive cancer cells (Supplementary Fig. 10a), suggesting that they were most likely generated via HIF1-mediated signals. On the other hand, CI-deficient tumor vasculature in both $143B^{-/-}$ and $HCT^{-/-}$ xenografts was characterized mainly by small, lumen-free and SMA-negative vessels (Fig. 5a), indicating an impairment in vascular maturation. The re-expression of NDUFS3 in $143B^{-/-}$ cells restored HIF-1α-associated vasculature (Supplementary Fig. 10b), whereas tumors in which CI was depleted by Dox treatment displayed a greater number of SMA-negative vessels than their untreated controls (Supplementary Fig. 10c), demonstrating that immature vasculature was a consequence of CI dysfunction. Moreover, complementing $143B^{-/-}$ and $HCT^{-/-}$ cells with TM-HIF-1α also rescued vessel maturation (Supplementary Fig. 10d), indicating that lack of an efficient vasculogenesis contributed to the lower tumorigenic potential of CI-deficient tumors. Since SMA is also a marker of cancer associated fibroblasts (CAFs), its staining showed that mature lumen-bearing vessels in $143B^{-/-}$ and $HCT^{-/-}$ xenografts were located almost exclusively in the murine stromal component of the tumor (Fig. 5a), an observation corroborated by staining collagen fibers (Fig. 5b). Furthermore, as we previously demonstrated, xenografts derived from cell lines carrying high loads (>85%) of CI disruptive m.3571insC/*MT-ND1* mtDNA mutation[19] also showed more pericyte-negative endothelial structures than their CI-competent controls and displayed lumen-bearing vessels mainly in the stromal component (Supplementary Fig. 10e). Overall, histology revealed that the stromal component is more prominent in CI-deficient than in CI-competent tumors, where it was located mainly on the tumor periphery (Fig. 5c). We next quantified the contribution of non-cancer cells in CI-deficient tumors at day 10 and day 30 post injection, to reconstruct the timeline of their recruitment. At day 30, corroborating the histology results, both

143B and HCT xenografts displayed higher stromal contribution in association with CI deficiency (Fig. 5d). These data, collected from both epithelial and mesenchymal tumors, and from different models of CI dysfunction, identified a more general rearrangement of tumor microenvironment in CI-deficient tumors, possibly involved in supporting their proliferation. In particular, since the stromal component of CI-deficient tumors was associated with their vasculature, we hypothesize that the microenvironment may be involved in compensatory mechanisms triggered to overcome the inability of CI-deficient tumors to activate HIF-1α-mediated angiogenesis.

**Macrophage infiltration is a hallmark of CI-deficient tumors.** The role of microenvironment in supporting tumor growth is well acknowledged, in particular as a source of pro-angiogenic factors[32]. However, the microenvironment component in tumors may be highly heterogeneous and exert both protumorigenic and anti-tumorigenic functions. Thus, to identify the components which might support CI-deficient tumor growth, we next characterized the contribution of immune cells (CD45 +) and fibroblasts (CD31−CD45−) (Supplementary Fig. 11a). At day 10, no difference was observed in 143B xenografts, whereas $HCT^{-/-}$ tumors showed an increased contribution of each of the populations analyzed (Supplementary Fig. 11b). At day 30, both 143B and HCT CI-deficient xenografts harbored more CD45 + cells than their CI-competent counterparts (Supplementary Fig. 11b), identifying immune cells abundance as a phenomenon shared in the CI-deficient tumor microenvironment. Thus, we further characterized the populations of the innate immune system in our models, namely macrophages (F4/80 + Lys6G−), neutrophils (Lys6G + F4/80−), natural killer cells (NK, CD49b +), and dendritic cells (CD11c + F4/80−) (Supplementary Fig. 11c). Of note, a higher number of necrosis-associated neutrophils was observed in CI-competent tumors (Fig. 6a and Supplementary Fig. 11d). The most striking and consistent difference we observed involved a higher number of macrophages in both 143B and HCT CI-deficient xenografts (Fig. 6a), particularly evident from the increase in the macrophage-to-cancer cells ratio (Fig. 6b). With the aim to investigate whether the higher macrophage abundance may be due to increased monocyte differentiation, we analyzed CD11b and Ly6C markers at day 30 (Supplementary Fig. 12a). Both tumor types harbored comparable numbers of classical CD11b + Ly6C + monocytes, whereas the number of differentiated CD11b + LyC− monocytes was higher in CI-deficient masses (Supplementary Fig. 12b). Indeed, the number of CD11b + Ly6C−F4/80 + cells was also higher in CI-deficient tumors (Fig. 6c), suggesting that CI deficiency in cancer may promote differentiation of CD11b + Ly6C + cells into macrophages (CD11b + Ly6C−F480 +). Strikingly, histology both at day 10 and at day 30 (Fig. 6c), revealed a difference in macrophage localization. While in CI-competent tumors they were restricted mainly at the tumor periphery, in CI-deficient tumors macrophages were found infiltrating the cancer mass (Fig. 6d). In line with this observation, we identified the down-regulation of macrophage migration inhibitory factor (MIF) in CI-deficient cancer cells (Supplementary Fig. 13a). Interestingly, MIF is a HIF1-induced protumorigenic cytokine whose down-regulation has recently been associated with macrophage-mediated angiogenesis[33]. Indeed, we corroborated reduced levels of MIF in $143B^{-/-}$ tumors and established that the complementation with TM-HIF-1α rescues MIF expression and reduces the number of macrophages in $143B^{-/-}$ tumors (Supplementary Fig. 13b–d), suggesting the increased presence of macrophages in CI-deficient tumors may be a consequence of HIF1 inactivation.

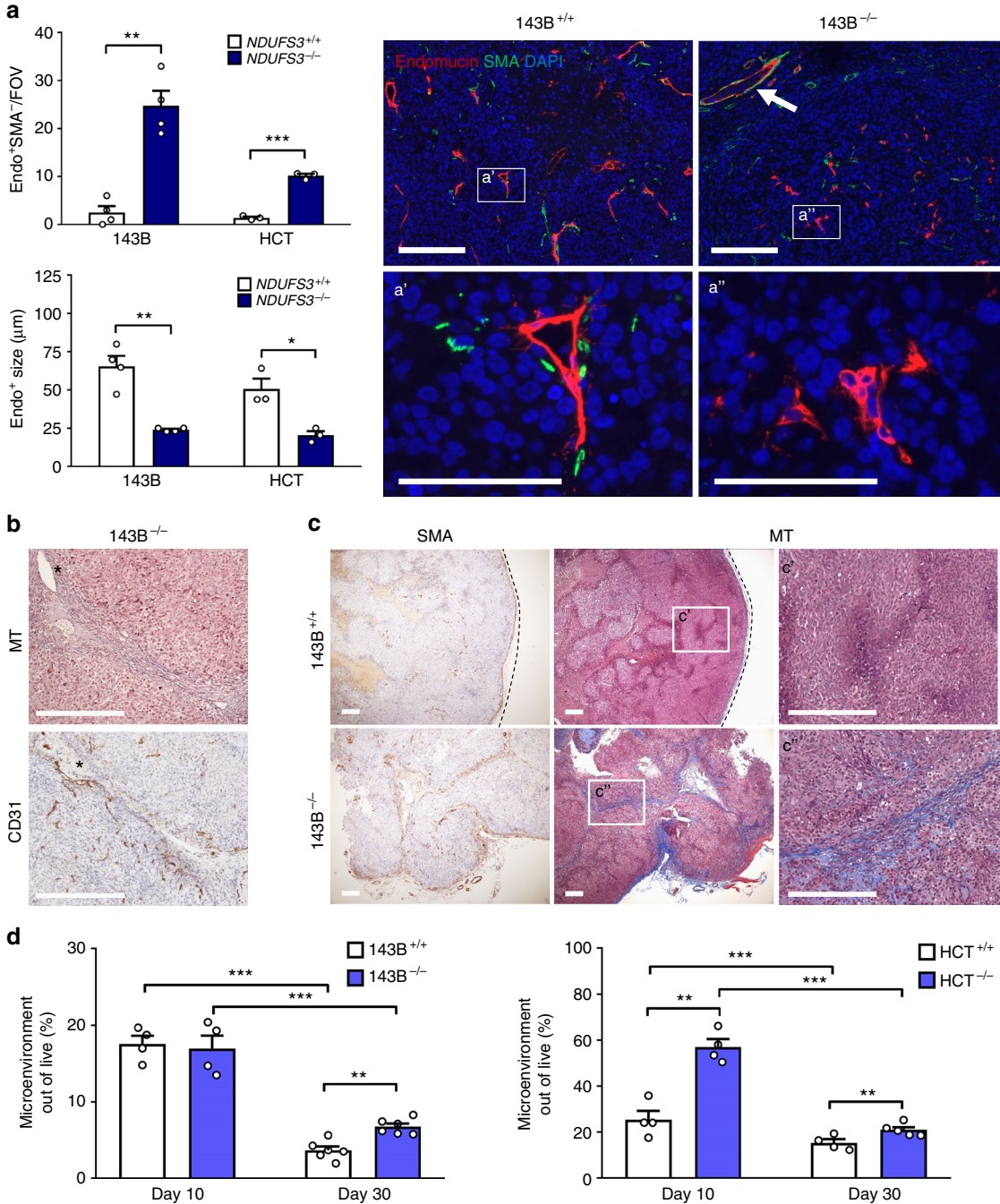

**Fig. 5** CI-deficient tumors display stroma-associated angiogenesis. **a** Number of pericyte-negative (Endo$^+$SMA$^-$) vessels per field of view (FOV) and average size of Endo$^+$ formations in 143B [$n = 4$, df(Endo$^+$SMA$^-$) = 6, $t$(Endo$^+$SMA$^-$) = 6.6, df(size) = 3, $t$(size) = 6] and HCT [$n = 3$, df = 4, $t$(Endo$^+$SMA$^-$) = 18.5, $t$(size) = 4.2] xenografts. For vessel size in 143B xenografts, the Student's $t$-test assuming unequal variance was applied. Representative images of immunofluorescent staining analyzing vessel morphology are shown. The arrow indicates a lumen-bearing vessel in the stromal area of a CI-deficient tumor. Scale bars: 100 μm. Data are mean + s.e.m. **b** Masson's trichrome (MT) and CD31 immunohistochemistry of CI-deficient xenografts. Representative images are shown. Scale bars: 100 μm. **c** Masson's trichrome (MT) and smooth muscle actin (SMA) staining in 143B$^{+/+}$ and 143B$^{-/-}$ xenografts. Dashed lines delineate the tumor margin. Scale bars: 100 μm. **d** Flow cytometry analysis of murine (stromal) cells in 143B and HCT xenografts at day 10 [$n = 4$, df = 6, $t$(143B) = 0.3, $t$(HCT) = 6.2] and day 30 [$n$(143B) = 6, df(143B) = 10, $t$(143B) = 4.8, $n$(HCT) = 4, df(HCT) = 6, $t$(HCT) = 2.9] post injection. Data are mean + s.e.m. In each panel, statistical significance is specified with asterisks (*$p < 0.05$, **$p < 0.01$, ***$p < 0.001$)

Since changes in the CAFs distribution were observed on histology (Fig. 5c and Fig. 6e), and since both CAFs and tumor associated macrophages (TAMs) have been shown to exert well-characterized pro-angiogenic roles[32,34], it is reasonable to hypothesize they might converge in promoting survival in CI-deficient tumors. However, in line with the flow cytometry data at day 30, the

number of single CAFs in the tumor was comparable between the two groups (Supplementary Fig. 14a), indicating that CI-competent and deficient cells were equally able to activate fibroblasts, which was also confirmed in vitro (Supplementary Fig. 14b). On the other hand, the number of single macrophages was higher in CI-deficient tumors than in control masses (Fig. 6d), suggesting that

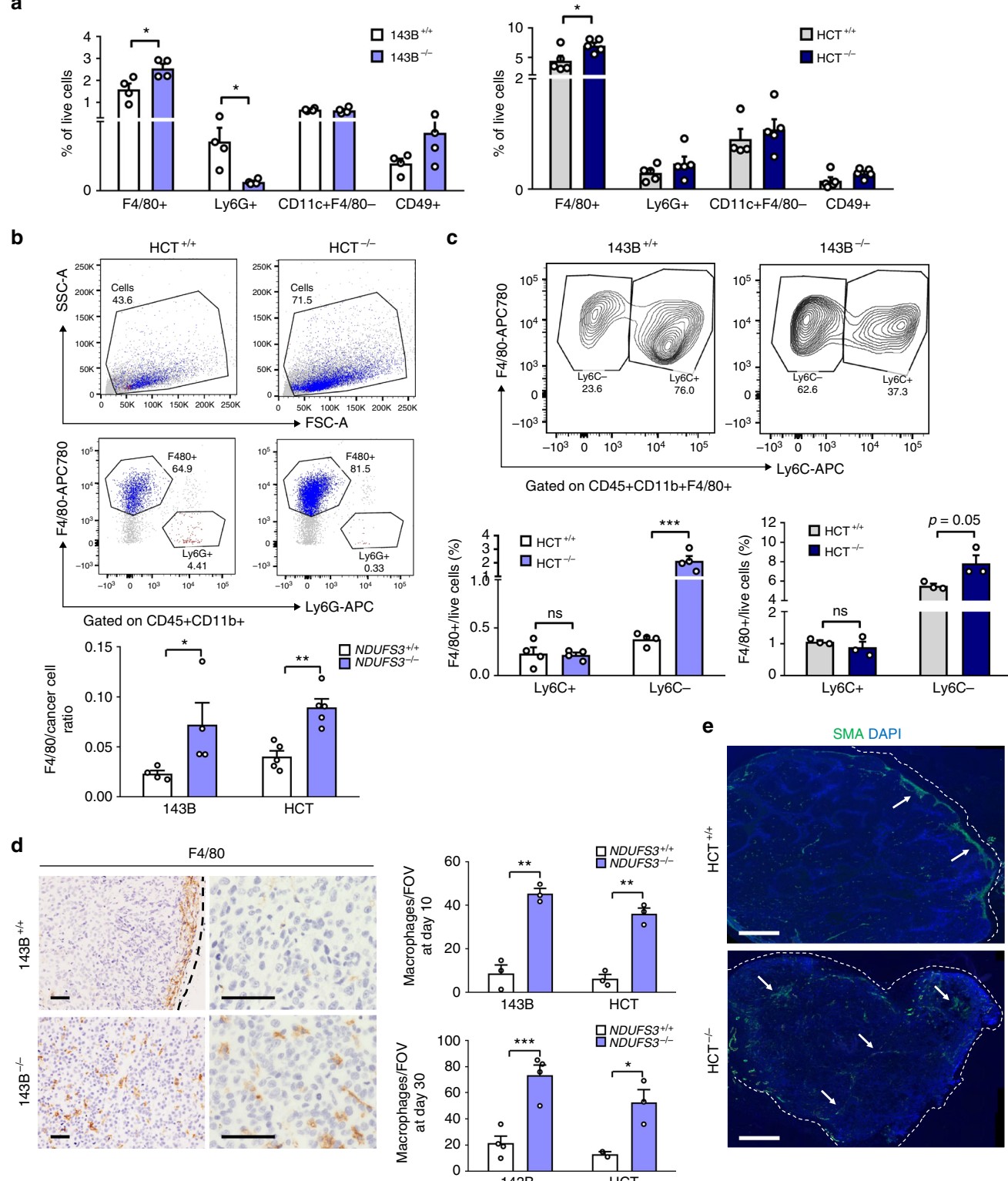

macrophages might contribute more than CAFs in progression of CI-deficient tumors. The prominent contribution of macrophages in association with CI-deficient cancer cells was furthermore observed in tumors in which the CI defect was due to high mutant loads of the m.3571insC/MT-ND1 mutation (Supplementary Fig. 15a), as well as in *Drosophila lgl*[−/−]*Ras*[V12]*V1*[KD] tumors where the macrophage component (NimC1+) of the hemocytes (the fly blood cells) was found increased compared to their controls

(Supplementary Fig. 15b). Additionally, the treatment of CI-competent xenografts with metformin, besides reducing tumor growth at comparable levels with genetic CI KO (Supplementary Fig. 15c), led to a higher number of intra-tumor macrophages than their untreated controls, as well as to a higher number of CD11b + F4/80 + Ly6C− cells (Supplementary Fig. 15d, e). Thus, our data imply that targeting CI results in an increase of macrophages within the tumor, possibly in order to support survival.

**Fig. 6** Macrophage infiltration is a hallmark of CI-deficient tumor microenvironment. **a** Flow cytometry analysis of innate immune system populations in 143B ($n = 4$, df = 6) and HCT ($n = 5$, df = 8) tumors. The contribution of macrophages (F4/80 +) [$t$(143B) = 2.8, $t$(HCT) = 2.8], neutrophils (Lys6G +) [$t$(143B, data were log-transformed) = 5], dendritic cells (CD11c + F4/80−) and natural killer cells (CD49b +) is shown at day 30 post injection in ICRF nude mice. Data are mean + s.e.m. **b** Macrophage to cancer cell ratio in 143B ($n = 4$, df = 6, $t = 3.21$) and HCT ($n = 5$, df = 8, $t = 4.834$) xenografts. Data are mean + s.e.m. Representative dot-plots display contribution of macrophages (F4/80 +, in blue) and neutrophils (Ly6G +, in red) among 100,000 acquired events (SSC-A/FSC-A plots) and their relative amount among CD45 + population (F4/80/Ly6G plots). All other cell populations are represented in gray. The gates indicate population percentages. **c** Flow cytometry analysis of undifferentiated (CD11b + Ly6C +) and differentiated (CD11b + Ly6C−) monocytes in 143B ($n = 4$, df = 6, $t = 4.213$) and HCT ($n = 3$, df = 4, $t = 2.674$) xenografts at day 30 post injection in ICRF nude mice. Representative contour-plots are shown. The gates indicate population percentages. Data are mean + s.e.m. **d** The number of macrophages (F4/80 +) infiltrating tumor tissue counted per field of view (FOV) in xenografts at day 10 [143B ($n = 3$, df = 4, $t = 8$), HCT ($n = 3$, df = 4, $t = 9$)] and at day 30 [143B ($n = 4$, df = 6, $t = 5.376$), HCT ($n = 3$, df = 3, $t = 3.201$)] post injection. Representative images of immunohistochemistry analysis for macrophage marker F4/80 in xenografts are shown. Dashed lines delineate the tumor margin. Scale bars: 50 μm. Data are mean + s.e.m. **e** Representative image of immunofluorescent staining of the SMA + microenvironment component in HCT xenografts. Dashed lines delineate the tumor margin. The arrows indicate the collective CAF formations. Scale bars: 50 μm. In each panel, statistical significance is specified with asterisks (*$p < 0.05$, **$p < 0.01$, ***$p < 0.001$)

**Inhibiting TAMs increases efficacy of CI targeting.** TAMs may exert either pro-inflammatory or pro-angiogenic functions during tumor progression, depending on whether they are polarized toward M1 or M2 population, respectively[35]. Since abnormal vascularization, comparable to the Endo + SMA− structures identified in CI-deficient cancers (Fig. 5a) has been associated with M2-mediated pro-angiogenic signals[36], we analyzed CI-competent and deficient xenografts for the markers generally associated with M1 (iNOS) or M2 (CD206, Arg1) macrophage populations. In all samples analyzed, the number of CD206-expressing macrophages increased from day 10 to day 30, when more than 70% of TAMs were CD206 +, whereas the number of macrophages expressing iNOS was generally low in all samples analyzed (10–20% in HCT and 1–4% in 143B), suggesting that the activation of protumorigenic TAMs is a property of both CI-deficient and control tumors (Supplementary Fig. 16a). Interestingly, iNOS expression, usually associated to M1 macrophages, was observed in CD206$^{high}$ and Arg1$^{high}$ population (Fig. 7a). There was no increase in the relative expression of any of the markers in CI-deficient tumors (Supplementary Fig. 16b), excluding the possibility that CI-deficient cancer cells are more capable to polarize TAMs to M2. Nonetheless, the total number of CD206 + and Arg1 expressing macrophages was higher in 143B$^{−/−}$ and HCT$^{−/−}$ masses, whereas the number of iNOS expressing TAMs was either comparable or higher in CI-competent tumors (Fig. 7b), showing that protumorigenic macrophages were generally more abundant in CI-deficient tumors. These data demonstrate that targeting CI did not potentiate M2 polarization, but was instead associated with higher macrophage abundance, in agreement with the increase in monocyte differentiation and higher macrophage infiltration identified in CI-deficient tumors (Fig. 6b–d).

Collectively, our data suggest that CI-deficient xenografts could rely on TAMs to promote survival. To prove this hypothesis, we depleted macrophages during CI-deficient tumor development. HCT$^{−/−}$ xenografts were treated with liposomes containing clodronate, a bisphosphonate that induces selective apoptosis of phagocyting macrophages. Compared to vehicle-treated controls, mice receiving clodronate showed almost complete absence of xenograft growth (Fig. 7c). Despite the anti-tumorigenic effect of clodronate was observed also in HCT$^{+/+}$ tumors (Supplementary Fig. 17a), the treatment did not prevent HCT$^{+/+}$ cells from forming frank tumors, with histology similar to the untreated controls (Supplementary Fig. 17b). Instead, HCT$^{−/−}$ xenografts treated with clodronate formed masses not larger than 10 mm$^3$, which upon excision revealed to be white, indicating complete absence of vascularization (Fig. 7c). Similarly, when 143B$^{−/−}$ cells were injected in more permissive $Rag1^{−/−}FVB/n$ mice, clodronate-treated tumors presented with a lower number of

SMA-negative vessels (Supplementary Fig. 17c, d). Importantly, histology of clodronate-treated HCT$^{−/−}$ masses revealed extensive necrosis, normally absent from CI-deficient xenografts (Fig. 7c), demonstrating that CI-deficient cells depend on macrophages to form tumors. This finding has the potential clinical implication of enhancing metformin treatment in cancer, if combined with drugs blocking macrophages activity. Thus, we treated HCT$^{+/+}$ xenografts simultaneously with both metformin and colony stimulating factor 1 (CSF1) receptor inhibitor PLX-3397, with the aim to specifically block CSF1-dependent macrophage infiltration in tumors. In line with the data from CI-deficient models, the combination of drugs significantly increased the efficacy of both PLX-3397 and metformin alone (Fig. 7d), proving that targeting macrophages is a promising approach to potentiate the effects of CI inhibitors in cancer, in a combinatorial regimen.

## Discussion

Targeting CI is considered to be a promising anti-cancer therapeutic strategy, according to the bulk of data collected both in experimental settings[1,37,38] and in clinical trials using metformin[2,39]. This is indirectly supported by the negative selection of severe CI defects in human tumors, which in the rare occasions when they do occur, associate with the development of low-proliferative oncocytomas[8]. We here exploit a fine-tunable system of CI ablation in different cancer types to trigger conversion to oncocytomas, and demonstrate that HIF-1α destabilization is accountable for such indolent behavior. Nonetheless, the conflicting data regarding metformin efficacy[40–43] and the fact that CI-deficient lesions continue to strive, point out the triggering of adaptive responses which our models allowed to dissect. In particular, we identified TAM-associated survival response in CI-deficient tumors and demonstrated that macrophage inhibition further decreased their tumorigenic potential.

Although HIF-1α destabilization has been previously described in CI-deficient cancer models, human oncocytomas, as well as in xenografts treated with metformin[1,10,19], the molecular mechanisms linking these phenomena are still unclear. In particular, at least two players undergo significant changes in cancer cells deprived of CI, converging toward the promotion of PHD activity and HIF-1α destabilization regardless of hypoxia, one of them being the accumulation of α-KG[30]. Further, our results support the concept that OXPHOS alterations cause the redistribution of intracellular O$_2$ from respiratory enzymes to PHDs[31]. In this metabolic frame of HIF-1α regulation, α-KG and O$_2$ likely decrease the ability of CI-deficient cells to sense hypoxia, which is reflected in the hypoxic foci lacking nuclear HIF1 in CI-deficient tumors.

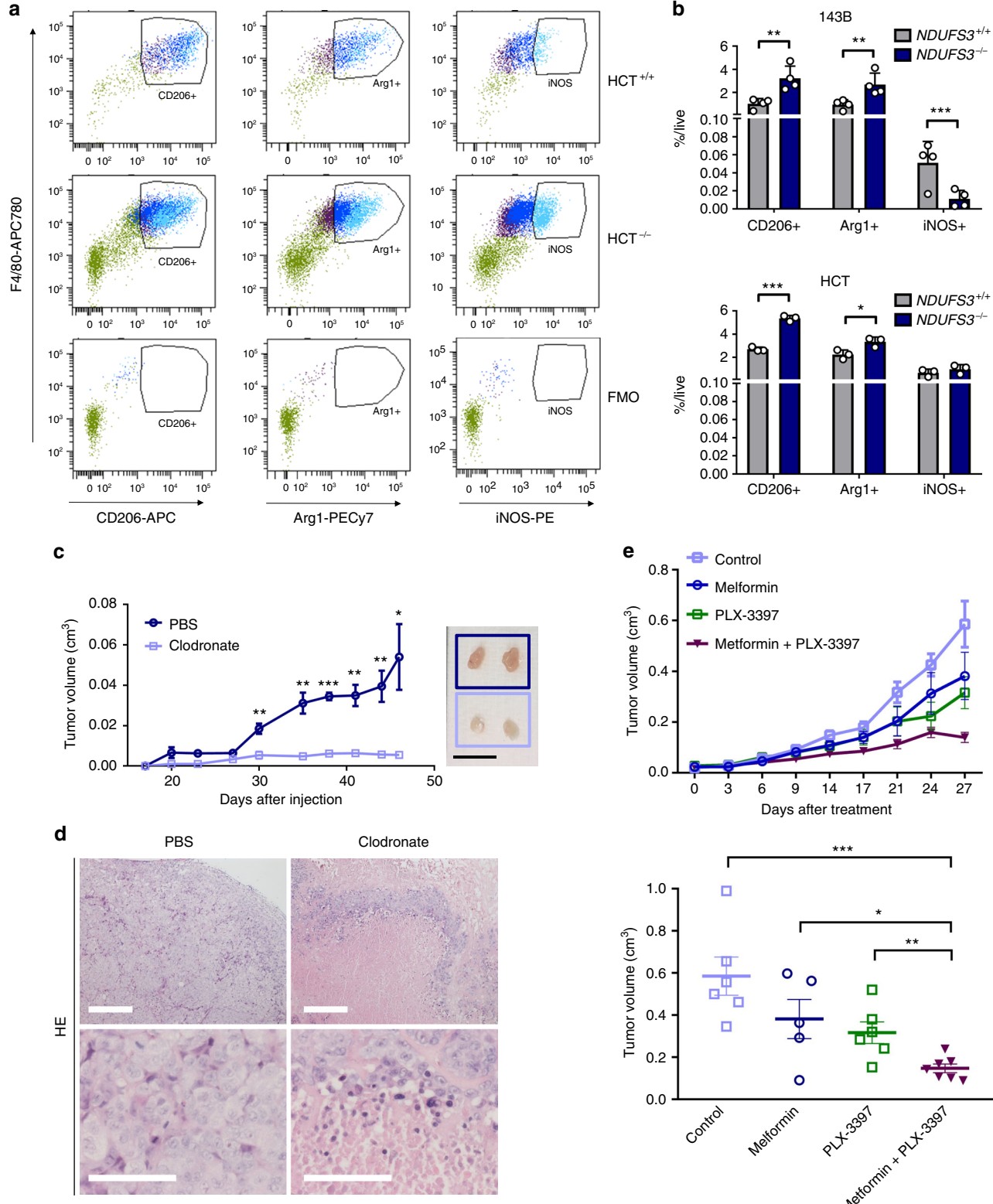

Due to the great number of pro-survival mechanisms activated by HIF1, it is reasonable to imply that the loss of HIF-1α challenges progression of CI-deficient masses. By complementing CI-deficient cells with TM-HIF-1α, we have provided proof for such concept. The consequences of HIF-1α destabilization in CI-deficient tumors are at least dual. As previously shown, it prevents the acquisition of a typical Warburg profile[20], which is linked to the upregulation of several pro-glycolytic genes HIF1

is responsible to transcribe. More importantly, as demonstrated here, HIF1 inactivation impairs tumor vasculature in CI-null tumors, compromising their nutrient supply.

However, despite the loss of HIF-1α and maintenance of a low-proliferative phenotype, CI-deficient tumor growth persists. In this respect, our observation of a different histological architecture in CI-deficient masses revealed that the striking abundance of microenvironment component in these tumors may be

**Fig. 7** TAMs support CI-deficient tumors growth and their inhibition enhances metformin effects. **a** Representative dot plots displaying macrophage populations expressing CD206, Arg1, and iNOS in $HCT^{+/+}$ and $HCT^{-/-}$ xenografts, and in fluorescence minus one (FMO) controls. iNOS (light blue) was set as the first front and Arg1 (dark blue) as the second front population, compared to CD206 (dark purple) and remaining immune cells (green). **b** Flow cytometry analysis of macrophage subpopulations markers CD206, Arg1, and iNOS in 143B and HCT tumors at day 30 post injection in ICRF nude mice. Statistics for 143B: $n = 4$, $df = 6$, $t(CD206) = 3.756$, $t(Arg1) = 3.219$, $t(iNOS) = 3.143$. Statistics for HCT: $n = 3$, $df = 4$, $t(CD206) = 14.96$, $t(Arg1) = 3.348$. Data are mean + s.e.m. **c** Tumor growth curves of $HCT^{-/-}$ xenografts in ICRF nude mice treated with or without clodronate [$n = 8$, $df = 14$, $t(day 47) = 3.524$]. Representative tumors are shown. Scale bars: 1 cm. Data are mean ± s.e.m. **d** Hematoxylin/eosin (HE) staining of $HCT^{-/-}$ xenografts in ICRF nude mice treated with or without clodronate. Scale bars: 50 μm. **e** Tumor growth curves and volume at day 27 post-treatment of $HCT^{-/-}$ xenografts in CD-1 nude mice, treated with metformin (2 mg mL$^{-1}$ in drinking water) or PLX-3397 (1.5 mg per day by oral gavage) alone and in combination [data were log-transformed, $n = 5$–7, df(metformin versus double treatment) = 10, $t$(metformin versus double treatment) = 2.539]. Data are mean ± s.e.m. In each panel, statistical significance is specified with asterisks (*$p < 0.05$, **$p < 0.01$, ***$p = 0.001$)

accountable for their alternative, HIF-1α-independent nutrient supply. Even though this needs to be corroborated in a fully immunocompetent system, it is well-known that cells from the tumor microenvironment promote its progression[44], and that the ultimate vascular phenotype of a neoplasm results from converging/synergistic cancer cell-autonomous and microenvironment-mediated pro-angiogenic signals[32]. Hence, abnormalized vessels typical of myeloid-driven vasculogenesis[45], the infiltration of macrophages observed in CI-deficient tumors and the reduction of their number when HIF-1α activity was complemented by TM-HIF-1α, all together imply that macrophage abundance is a consequence of HIF-1α destabilization in CI-deficient tumors. This is furthermore supported by the fact that CI-deficient tumors display MIF downregulation, a phenomenon recently associated with macrophage-mediated vascularization activated as a compensatory response upon anti-VEGF treatment[33]. Indeed, since MIF is a known HIF1-responsive gene and CI targeting ultimately leads to lack of HIF1, our data indicate that TAM abundance may be a consequence of MIF downregulation. Intriguingly, MIF is a classical pro-survival cytokine, meaning that its downregulation may also contribute to the lower tumorigenic potential of the CI-deficient cancer cells. However, at the same time its downregulation paradoxically releases a microenvironment-mediated compensatory response, classifying MIF in the ever-growing group of double-edged oncojanus cancer genes[46]. Considering the complexity of the cross-talk between cancer cells and microenvironment, we acknowledge that mechanisms apart from the HIF–MIF axis likely contribute to macrophage abundance in CI-deficient tumors. The role of CAFs needs to be further explored, since their secretion of extracellular matrix proteins supports macrophage migration, and CAFs also secrete their own pro-angiogenic factors[32]. It is interesting to note that repair-like patrolling macrophages (CD11b + Ly6C −F4/80 +), we here identify as associated with CI-deficient cancers, have been described to promote angiogenesis and fibrosis in the wound[47], meaning that their abundance in the context of cancer might explain the fibroblast contribution we observe in CI-deficient tumors.

Blocking TAM infiltration further reduced the tumorigenic potential of CI-deficient xenografts, as well as metformin-treated tumors, indicating that simultaneous targeting of CI and macrophages is an efficient synergistic approach to reduce tumor growth. Of note, CI inhibitors such as metformin and macrophage-targeting agents both exhibit cytostatic effects on cancer cells. Confining cancers in a low-proliferative state ought to be considered a valid alternative approach to cancer eradication, since turning cancer into a chronic disease slows down its genetic evolution and allows the pool of progressing cancer cells to competitively consume available resources[48]. Interestingly, both metformin and macrophage inhibitors are economical drugs, currently used in the clinics, with limited side effects when administered in diabetic and osteoporotic patients, respectively.

Overall, our results corroborate targeting CI as a valid anti-cancer strategy, although this may be overcome, at least in certain types of cancers, by macrophage-mediated adaptive response. We hence look forward to the clinical trials exploring the combinatorial effects of drugs that hit two such essential players of the adaptive process of cancer cells.

## Methods

**Cell lines and treatments.** Osteosarcoma 143B Tk⁻ cells and colorectal cancer HCT116 cells were used, both carrying wild-type mtDNA. 143B cells were purchased from ATCC (#CRL-8303) and HCT116 cells were a kind gift from Prof. Paolo Pinton from the University of Ferrara. Cell origin was authenticated using AMPFISTRIdentifiler kit (Applied Biosystems #4322288) and their STR profile corresponded to their putative background (Supplementary Fig. 1a). For basal conditions, cells were cultivated in Dulbecco's modified Eagle medium (DMEM) high glucose (Euroclone #ECM0749L), supplemented with 10% FBS (Euroclone #ECS0180L), L-glutamine (2 mM, Euroclone #ECB3000D), penicillin/streptomycin (1 ×, Euroclone #ECB3001D), and uridine (50 μg mL$^{-1}$, Sigma-Aldrich #U3003), in an incubator with a humidified atmosphere at 5% $CO_2$ and 37 °C. Cells were replaced by a fresh batch after 15 passages and mycoplasma testing was performed before disposal and after each thawing (approximately every 2 months). Experiments in hypoxia were performed using an Invivo2 300 (Baker Ruskinn) chamber, set at 5% $CO_2$, 37 °C and 1% $O_2$. Where indicated, cells were incubated for 3 h with dimethyloxalylglycine [DMOG (1 mM), Sigma-Aldrich #D3695] or MG132 (10 μM, Sigma-Aldrich #M7449) and 1 h with pimonidazole (100 μM, Hypoxiprobe #70132-50-3).

**Genome editing for generation of NDUFS3 knockout.** Plasmids containing cDNA of *NDUFS3*-targeted zinc finger endonucleases were purchased from Sigma-Aldrich (#CKOZFND15168) and were used according to the manufacturer's instructions. Briefly, the plasmids were purified and transcribed in vitro using MessageMax caping kit (Cell Script #C-MMA60710), Poly adenylation kit (Epicentre #PAP5104H), and purified using MegaClear kit (Life Technologies #AM1908). The pool of zinc finger endonucleases mRNAs (2.5 μg each) was transfected using Transit-mRNA transfection kit (Mirus #2225) into 70% confluent cells. Cells were split 48 h after transfection and DNA was extracted using Mammalian Genomic DNA Miniprep Kit (Sigma-Aldrich #G1N350). Non-homologous repair efficiency was evaluated by Fluorescent PCR using KAPA2G Taq polymerase (Kapa Biosystems #KK5601) with 58 °C annealing and primers forward [Flc] CTGCCACAAGGAGCTAGGAC and reverse GCACAGGGAGATAAAAGGCA. Clonal selection was performed in order to identify the cells with frameshift *NDFUS3* mutations. Selection media used for the single-cell growth in 96-well plates was composed of DMEM high glucose (Euroclone #ECM0749L), supplemented with 20% FBS (Euroclone #ECS0180L), L-glutamine (2 mM, Euroclone #ECB3000D), penicillin/streptomycin (1 ×, Euroclone #ECB3001D), uridine (50 μg mL$^{-1}$, Sigma-Aldrich #U3003), L-Tryptophan (32 mg/L, Sigma-Aldrich #T0254), and Nicotinamide (8 mg L$^{-1}$, Sigma-Aldrich #72340). DNA extraction from 96-well plates was performed using 8 μL of Lysis Solution (Sigma-Aldrich #L3289) and 80 μL of Neutralization Buffer (Sigma-Aldrich #N9784) per sample. Upon genotyping, heterozygous clones were first expanded, and subjected to a subsequent second transfection with NDUFS3 zinc finger pool. This step led to the selection of homozygous frameshift *NDUFS3* mutants and homozygous wild-type revertants (Supplementary Fig. 1b). The latter were used as isogenic controls in all experiments described here. *NDUFS3* genotype was confirmed by Sanger sequencing using KAPA2G Taq polymerase (Kapa Biosystems #KK5601) and Big Dye protocol (Life Technologies #4337451). After the validation of single NDUFS3$^{-/-}$ clones, they were pooled for the experiments that followed (143B $n = 5$, HCT $n = 3$).

**Generation of cells with inducible NDUFS3 knockout.** To create a stable transgenic cell line that re-expresses NDUFS3 and allows its inducible knockout, the Retro-X Tet-Off Advanced Inducible Expression system (Clontech #632105)

was used by following manufacturer's instructions. First, $3 \times 10^6$ amphotrophic Phoenix (phxA) cells (ATCC® No. CRL-3213™) were transiently transfected with 20 µg of pRetroX-Tet-Off Advanced vector and 0.9 µg of helper plasmid using Lipofectamine 2000 (Thermo Fisher #11668027). PhxA cells were maintained in the presence of transfection mix for 16 h, then switched to DMEM for 24 h, after which the retroviral supernatant was collected, filtered (0.45 µm), supplemented with 8 µg mL$^{-1}$ hexadimethrine bromide, and 2 mL was used to stably transfect 143B cells ($50 \times 10^3$) in 6-well plates. The later were then (i) centrifuged 1800 r.p.m. at 32 °C for 45 min and incubated at 32 °C for 2 h, (ii) replenished with fresh retroviral supernatants, re-centrifuged as indicated and incubated at 32 °C for additional 4 h, (iii) replenished with normal medium and incubated overnight, (iv) replenished with 4 mL of fresh retroviral supernatant, centrifuged as indicated and incubated at 32 °C for 5 h, (v) replenished with normal medium and incubated 48 h, (vi) replenished with normal medium containing G418 (400 µg mL$^{-1}$, Sigma #A1720) and incubated for 48 h. The surviving cells were maintained in medium supplemented with 100 µg mL$^{-1}$ G418. Single-cell cloning was then performed and luciferase was transfected in each clone in order to identify cells with optimal transactivator activity. In particular, 2 µg of pRetroX-Tight-Pur-Luc Control Vector was transiently transfected using X-tremeGENE HP DNA Transfection Reagent kit (Roche #06366236001) in 143B cells ($80 \times 10^3$) seeded in 6-well plates in the standard medium but containing Clontech tetracycline-free FBS. Each clone was then treated with/without Dox (100 ng mL$^{-1}$, Sigma #D9891) and after 48 h, cells were harvested and processed according to the Dual-Luciferase Reported Assay System (Promega #E1910) protocol. The luciferase activity was generally high in absence of Dox and low in cells cultivated with Dox medium. Five clones with lowest + Dox luciferase activity were pooled together and then the second viral transduction was performed as described above, using pRetroX-Tight-Pur-NDUFS3 vector. The selection of cells that uptaken the plasmid was performed using puromycin (0.5 µg mL$^{-1}$, Sigma #P8833). Single-cell cloning was performed to identify clones showing lack of NDUFS3 after Dox treatment and five clones were pooled to obtain 143B$^{-/-NDUFS3}$ cell line.

**Generation of cells carrying TM-HIF-1α**. Wild-type sequence of *HIF1A* cDNA was cloned in pGEM (Promega #A1360) vector and mutagenesis was performed using Quickchange Site-Directed Mutagenesis kit (Agilent #200518), following manufacturer's instructions, to induce the mutations at the PHD hydroxylation sites following previously described indications[24]. In particular, three mutations were inserted in the *HIF1A* sequence, to substitute the prolyl hydroxylase (PHD)-targeted prolines and Factor Inhibiting HIF (FIH)-targeted asparagine with residues that cannot be hydroxylated (P402A, P564G, and N803A). The triple mutant *HIF1A* was then transferred to pMSCV-Puro retroviral vector (Clontech #PT3303-5). The empty vector and the vector containing TM-HIF-1α were then used to transduce 143B and HCT116 cells by following phxA transduction protocol described above. Cells carrying the vectors were selected with 2 µg mL$^{-1}$ puromycin (Sigma #P8833) and maintained in medium supplemented with 1 µg mL$^{-1}$ puromycin. Clonal selection was performed to identify the clones with the highest TM-HIF-1α protein levels in normoxia and a pool of 20 clones was made.

**SDS-PAGE and western blot**. Whole lysates of cultured cells and freshly snap-frozen xenograft samples were prepared in RIPA buffer (Tris–HCl pH 7.4 (50 mM), NaCl (150 mM), SDS (1%), Triton (1%), EDTA pH 7.6 (1 mM)) supplemented with protease inhibitors (Roche #11873580001), and quantified by Lowry protein assay (Bio-Rad #5000116). Samples were separated by SDS-PAGE and transferred onto nitrocellulose membrane using Turbo-pack system (Bio-Rad #1704159SP5). Membranes were blocked 30 min at 37 °C and incubated with primary antibodies using following dilutions/conditions: anti-NDUFS3 (AbCam #177471) 1:1000/1-h at room temperature (RT); anti-HIF1α (GeneTex #GTX127309) 1:2000/1-h at RT; anti-Vinculin (Sigma-Aldrich #V9131) 1:10,000/1-h at RT; anti-pimonidazole (Hydroxyprobe #4.3.11.3) 1:3000/2-h at 37 °C; anti-HIF-1α-OH (Cell Signaling #3434) 1:500/O/N at 4 °C; anti-VDAC (AbCam #154856) 1000/2-h at RT; anti-MT-CO2 (AbCam #110258) 1:1000/2-h at RT; anti-MIF (AbCam #175189) 1:5000/1-h at RT; anti-ACTB (Santa Cruz #SC-1615) 1:500/1-h at RT; anti-cleaved Caspase 3 (Cell Signaling Technology #9661) 1:1000/O/N at 4 °C; anti-PHD1 (Abcam #ab108980) 1:1000/O/N at 4 °C, anti-EGLN1/PHD2 (Novus Bio #NB100-137) 1:1000/O/N at 4 °C; anti-GAPDH (Sigma-Aldrich #G8795) 1:20,000/2-h at RT. Washes were performed 4 × 5 min using TBS-Tween [0.1% Tween 20 (Sigma-Aldrich #P9416) in Tris Buffered Saline] and incubation with secondary antibodies (Jackson ImmunoResearch Laboratories #111035144 and #111035146), diluted 1:20,000 in TBS-Tween, was performed for 30 min at RT. Developing was performed by using Clarity Western ECL Substrate (Bio-Rad #1705061) and exposing with ChemiDoc XRS$^+$ (Bio-Rad). For anti-HIF-1α-OH antibody, Western Breeze system (Life Technologies #WB7106) was used for secondary antibody and developing solutions, following the manufacturer's instructions. Raw acquisition images are presented in Supplementary Fig. 18.

**Mitochondrial-enriched fraction preparation**. Mitochondria-enriched fractions were obtained by subcellular fractionation ($5–10 \times 10^6$ cells) in presence of digitonin (50 µg mL$^{-1}$). Crude mitochondria were obtained from $15–20 \times 10^6$ cells. The cell pellet was suspended in ice-cold buffer containing 200 mM mannitol,

70 mM sucrose, 1 mM EGTA, 10 mM HEPES (pH 7.6) and mechanically disrupted with a glass/teflon Potter-Elvehjem homogenizer. Differential centrifugations ($600 \times g$ for 10 min at 4 °C followed by $14,000 \times g$ for 10 min at 4 °C) were performed to separate crude mitochondria from other subcellular fractions. Samples were stored at −80 °C.

**High-resolution clear native PAGE (hrCNE)**. For complex I in-gel activity (CI-IGA) assay and western blotting analysis, mitochondrial-enriched fractions were resuspended in mitochondrial buffer (750 mM aminocaproic acid, 50 mM Bis-Tris, pH = 7) and solubilized by adding DDM/protein ratio of 2.5 (g/g). Suspension was incubated on ice for 10 min and then centrifuged at $13,000 \times g$ for 15 min. Aliquots of supernatants (80 µg protein) were separated by 4–16% first dimension hrCNE gradient gel (NativePAGE™ 4–16% Bis-Tris Protein Gels, Invitrogen, #BN1002BOX) using as anode buffer 25 mM pH = 7 and as cathode buffer 50 mM Tris, 7.5 mM Imidazole, 0.02% n-Dodecyl β-D-maltoside (Sigma-Aldrich #D4641), 0.05% sodium deoxycholate (Sigma-Aldrich, #D6750). Proteins were either transferred onto a nitrocellulose membrane at 100 V for 1 h at room temperature or complex I in-gel activity (CI-IGA) assay was performed. Briefly, gel was rinsed in cathode buffer and incubated at room temperature for 15 min in a solution containing 2 mM Tris pH = 7.4, 0.5% 3-(4,5-Dimethyl-2-thiazolyl)-2,5-diphenyl-2H-tetrazolium bromide (MTT, Sigma-Aldrich, #M5655), 0.02% reduced β-Nicotinamide adenine dinucleotide (NADH, Sigma-Aldrich, #N8129) under constant agitation and protected from light.

**Measurement of ATP synthesis rate**. The rate of mitochondrial ATP synthesis driven by CI, CII, and CIII was measured in digitonin-permeabilized cells. After trypsinization, cells ($10 \times 10^6$ mL$^{-1}$) were suspended in a buffer containing 150 mM KCl, 25 mM Tris–HCl, 2 mM EDTA (ethylenediaminetetraacetic acid), 0.1% bovine serum albumin, 10 mM potassium phosphate, 0.1 mM MgCl$_2$, pH 7.4, kept at room temperature for 15 min, then incubated with 50 µg mL$^{-1}$ digitonin until 90–100% of cells were positive to Trypan Blue staining. Aliquots of $3 \times 10^5$ permeabilized cells were incubated in the same buffer in the presence of the adenylate kinase inhibitor P$^1$,P$^5$-di(adenosine-5′) pentaphosphate (0.1 mM) and OXPHOS complexes substrates, chemiluminescence was determined as a function of time with Sirius L Tube luminometer (Titertek-Berthold, Pforzheim, Germany). The chemiluminescence signal was calibrated with an internal ATP standard after the addition of 10 µM oligomycin. The rates of the ATP synthesis were normalized to protein content and citrate synthase (CS) activity[49].

**Glucose consumption**. Glucose consumption was determined by using the glucose oxidase (GO) assay kit (Sigma-Aldrich #GAGO20) scaling down the manufacturer protocol to a final volume of 1 mL. Briefly, cells were seeded in 6-well plates ($1 \times 10^5$ cells/well) in high glucose medium. After 48 h, cells were washed in PBS and incubated with DMEM-high glucose without phenol red. Aliquots of 100 µL of medium were taken at time 0 and after 24 h and 48 h of incubation and 3 µL of samples were used to determine the glucose concentration, by the enzymatic reaction of glucose oxidase coupled to peroxidase-mediated oxidation of reduced o-Dianisidine ($\lambda = 540$ nm, 30 min, 37 °C). The data obtained were normalized on cell number.

**Lactate production**. Cells were seeded in 6-well plates ($3 \times 10^5$ cells/well) in 2 mL of high-glucose medium. After 24 h and 48 h, aliquots of medium were collected and de-proteinated with 6% perchloric acid, vortexed and incubated in ice for 1 min. Samples were centrifuged at 13,000 r.p.m. at 4 °C for 2 min and the lactate concentration in supernatants was determined by measuring NADH ($\lambda = 340$ nm; $\varepsilon = 6.22$ mM$^{-1}$ cm$^{-1}$) production in a buffer containing 320 mM glycine, 320 mM hydrazine, 2.4 mM NAD$^+$ and 2 U mL$^{-1}$ L-lactic dehydrogenase (Sigma-Aldrich #L1006) after 30 min of reaction at 37 °C. Data were expressed as pmoles of lactate produced per cell.

**Oxygen consumption rate**. Mitochondrial respiration was evaluated using the Seahorse XFe Cell Mito Stress Test Kit (Seahorse Bioscience #103015-100) following the manufacturer instructions. Cells were seeded ($3 \times 10^4$ cells/well) into XFe24 cell culture plate and allowed to attach for 24 h. Cell culture media was replaced with XF media (Seahorse Bioscience #103334-100). OCR was measured over a 3 min period, followed by 3 min of mixing and re-oxygenation of the media. For Mito Stress Test, complete growth medium was replaced with 670 µL of unbuffered XF media supplemented with 10 mM glucose pH 7.4 pre-warmed at 37 °C. Cells were incubated at 37 °C for 30 min to allow temperature and pH equilibration. After an OCR baseline measurement, 70 µL of oligomycin, carbonyl cyanide-p-trifluoromethoxyphenylhydrazone (FCCP), and rotenone plus antimycin A were sequentially added to each well to reach final concentrations of 1 µM oligomycin, 0.25 µM FCCP, and 1 µM rotenone and antimycin A. Three measurements of OCR were obtained following injection of each drug and drug concentrations optimized on cell lines prior to experiments. At the end of each experiment, the medium was removed and SRB assay was performed to determine the amount of total cell proteins as described above. OCR data were normalized to total protein levels (SRB protein assay) in each well. Each cell line was represented

in five wells per experiment ($n = 3$ replicate experiments). Data are normalized on SRB absorbance and expressed as pmoles of $O_2$ per minute.

**ATP content measurement.** Total ATP content was determined by using the ATPlite Luminescence Assay System (Perkin Elmer #6016943). Cells ($3-1.5 \times 10^4$) were seeded in a 96-wells plate in high glucose DMEM. After 24 h, cells were washed twice with PBS and incubated with high glucose DMEM or with galactose DMEM. ATP content was measured after 16 h following the manufacturer protocol. Luminescence was detected using a multilabel counter Victor3 (Perkin Elmer). Data are normalized on cell number detected by SRB assay.

**Evaluation of Krebs-cycle metabolite concentrations.** Absolute metabolite concentrations were measured with Carcinoscope analysis [Human Metabolome Technologies (HMT)] that uses capillary electrophoresis coupled to time of flight/triple quadrupole mass spectrometry. Metabolites were extracted using 100% methanol supplemented with 550 μL of internal standard solution provided by HMT from 2 to 5 million cells seeded on 90 mm plate. The metabolite concentrations of in vitro cell preparations were normalized to the number of viable cells. Tumor samples were grinded in liquid nitrogen before proceeding with the analysis. The measurement was corroborated by LC-based metabolite quantification at the Metabolomics Core Technology Platform of the Excellence cluster CellNetworks (University of Heidelberg; grant no. ZUK 49/2010-3002962).

**Citrate $^{13}$C isotopomer analysis.** For mass isotopomer analysis, cells were incubated for 3 h either with DMEM high glucose (Euroclone #ECM0749L) supplemented with [$^{13}$C] labeled glutamine (2 mM) or with DMEM (Life Technologies #11966-025) supplemented with sodium pyruvate (110 mg L$^{-1}$, Sigma-Aldrich #P2256) and [$^{13}$C] labeled glucose (25 mM). Nutrients labeled with $^{13}$C were purchased from Cambridge Isotope Laboratories (#CLM-1396 and #CLM-1822). Metabolite extraction was performed as described above and the isotopomer distribution was evaluated with F-Scope analysis (Human Metabolite Technologies).

**Reactive oxygen species measurement.** To determine the $H_2O_2$ production, cells were incubated with 2 μM 2,7-dichlorodihydrofluorescein diacetate (H$_2$DCFDA) (Life Technologies #D399) added to 100,000 cells, for 30 min at 37 °C. The cells were then collected and the reaction was stopped by placing them in an ice bath for 5 min. The cells were disrupted by treatment with Triton X-100 (2%) and centrifuged at 2500 $\times g$ for 20 min at 4 °C. The supernatant was used to measure fluorescence emission (excitation, 485 nm; emission, 535 nm) using a multilabel counter Victor3 (Perkin Elmer, Turku, Finland). The amount of $H_2O_2$ produced was calculated by using a standard curve of 2,7-DCFH$_2$ in which 1 μM of 2,7-DCFH$_2$ represented 1 μM of $H_2O_2$[50].

**3D colony growth assay.** Tumor cells were seeded in 24-well dishes at 500 cells/dish in 150 μL of 2:1 Rat tail collagen-I (Corning #354249) and Matrigel (Corning #356234), yielding a final collagen concentration of 4 mg mL$^{-1}$ and a final Matrigel concentration of 2 mg mL$^{-1}$. The wells were pre-coated with 50 μL of collagen/matrigel media. The colonies were maintained in DMEM in basal conditions and analyzed after 14 days. For data in Fig. 1b the invasive margins of the colonies were counted in 10 random colonies per sample ($n = 3$). For data in Supplementary Fig. 8c, colony number and average diameter were evaluated in 10 random colonies per sample ($n = 6$).

**Xenograft growth.** For in vivo studies, $nu/nu$ mice (CD-1® Nude Mouse Crl:CD1-Foxn1ⁿ) were purchased from Charles River Laboratories. Where indicated, ICRF nude or $Rag1^{-/-}FVB/n$ mice available at The Francis Crick Institute Biological Research Facility were used. We have complied with all relevant ethical regulations for animal testing and research. The animals were treated according to institutional guidelines and regulations at Paracelsus Medical University Salzburg, The Francis Crick Institute and University of Bologna. Respectively, the study received ethical approval from the Salzburg State Ethics Research Committee (20901TVG/112), UK Home Office (project license PPL number P83B37B3C) and Italian Ministry of Health (authorization code 437/2018-PR). Five to six-week-old female mice were subcutaneously injected with a 100 μL suspension of $5 \times 10^6$ cells in serum free medium and matrigel (Corning #356234) in the right flank of the animal. For experiments in Figs. 3g, 7c and Supplementary Fig. 17a–d, a bilateral inoculation was performed. Xenograft size was measured with a sliding caliper twice a week, according to the formula: volume = width × height × length/2. Mice were sacrificed either simultaneously, when the first xenograft reached 10% of animal weight (Fig. 1a, f, Fig. 3g, Fig. 7c, Supplementary Fig. 8d, Supplementary Fig. 15c and Supplementary Fig. 17a–d), or consecutively, when each animal reached xenograft volume corresponding to 10% of animal weight or met the termination criteria (Fig. 2a and Fig. 7e). For the Dox-induced experiment, 3% sucrose with or without Dox (1 mg mL$^{-1}$) was added into the drinking water of the mice, which were sequentially randomized in Dox-treated and control group when tumor would reach 450 mm³. For CSF1-R inhibition, PLX-3397 (Pexidartinib, Apex Bio-Technology #B5854) was dissolved in DMSO and suspension was made by dilution in the aqueous solution of 0.5% hydroxypropyl methyl cellulose (HPMC, Sigma

#H7509) and 1% polysorbate (PS80, Sigma #59924). The drug (1.5 mg per mouse) was administered by oral gavage for 20 consecutive days with 100 μL of suspension. Metformin was added to drinking water at concentration of 2 mg mL$^{-1}$. Fresh water/metformin pouches were provided twice weekly. The PLX-3397 and/or metformin treatment was started when a tumor would reach 50 mm³. Prior to the sacrifice (3 h before) all animals were injected intraperitoneally with pimonidazole (60 mg kg$^{-1}$, Hypoxyprobe #70132-50-3) diluted in saline solution. For the clodronate treatment experiments in Fig. 7c and Supplementary Fig. 17a–d, the animals were pre-injected intraperitoneally with PBS or clodronate liposomes (100 μL, ClodronateLiposomes, Liposoma BV) on the day prior to cell injection. On the day of tumor cell injection, $5 \times 10^6$ cells in growth factor reduced matrigel (100 μL) were injected subcutaneously, immediately followed by injection of 40 μL of PBS or clodronate liposomes at the same position. The mice continued to receive intraperitoneal injection of liposomes twice weekly (100 μL).

**Immunohistochemical staining.** The samples were formalin fixed following standard protocols. Tissue sections (4 μm) were deparaffinized in xylene, rehydrated in absolute 2-propanol followed by heat-induced epitope retrieval in TE-T buffer (10 mM Tris pH 8.0, 1 mM EDTA, 0.05% Tween 20) for 40 min at 95 °C and 20 min at RT. Sections were equilibrated with phosphate-buffered saline containing 0.5% Tween 20 (PBS-T pH 7.4). Primary antibodies were diluted in Antibody diluent with background reducing components (Dako #S3022) and incubated at RT for 30 min. Blocking, secondary antibodies staining and development were carried out using the Envision Detection System (Dako #K4007 and #K4011) according to the manufacturer's instructions. Slides were counterstained with hematoxylin. The following primary antibodies were used: rabbit monoclonal anti-NDUFS3 (1:200, Abcam #177471); rabbit polyclonal anti-HIF-1α (1:350, Sigma-Aldrich #HPA001275); mouse monoclonal anti-pimonidazole (1:400, Hypoxyprobe #Mab-4.3.11.3); rabbit monoclonal anti-CD-31 (1:50; Abcam #28364); mouse monoclonal anti-KI-67 (1:100, Dako #M7240); mouse monoclonal anti-MT-CO1 (1:1000, Abcam #14705); mouse monoclonal anti-NDUFS4 (1:1000, Abcam #55540), and rat monoclonal F4/80 (1:100, eBiosciences #14-4801). Neutrophil marker 2b10 antibody was developed in house at The Francis Crick Institute. For macrophage staining with anti-F4/80, trypsin-based antigen retrieval (0.05% trypsin in 0.1 mM Calcium chloride solution, pH 7.8) was performed for 30 min at 37 °C, and biotinylated goat anti-rat secondary antibody (Sigma-Aldrich #A9037) was used, together with VECTASTAIN ABC-HRP Kit (Vector Laboratories #PK-4005). Hematoxylin/eosin and Masson's trichrome staining were performed on 4 μm sections following standard protocols. For evaluation of KI-67 positive nuclei, cells were counted at ×20 magnification in 5–10 fields of view per tumor. Macrophages (F4/80 +) were counted at of ×20 magnification in three fields of view per tumor.

**Electron microscopy.** Samples were fixed using paraformaldehyde (2%) and glutaraldehyde (2.5%) in cacodylate buffer (0.1 M) for 24 h and then washed, kept in the cacodylate buffer at +4 °C, fixed with glutaraldehyde (2.5%) and 1% osmium tetroxide, dehydrated in alcohol and propylene oxide and embedded in Epon 812. Sections (1 μm) were stained with 1% toluidine blue for morphology control and electron microscopy area selection. The sections were observed with JEM-1011 transmission electron microscope (JEOL Ltd). At least two different areas were observed for each tumor. Mitochondrial morphology was evaluated by measuring short mitochondrial axis and counting cristae in 50 mitochondria per sample.

**Quantitative real-time PCR.** Gene expression was analyzed following minimal information for publication of Quantitative Real-Time PCR (qRT-PCR) Experiments guidelines[51]. In particular, RNA was extracted using Trizol (Life Technologies #15596018) for cell lines, and Mammalian Genomic DNA Miniprep Kit (Sigma-Aldrich #RTN70) for snap-frozen xenograft samples. High Capacity cDNA Reverse Transcription Kit (Applied Biosystems #4368814) was used for preparation of cDNA with random hexamers starting from 300 ng of RNA. Primer sequences were designed using Primer3 software[52]. The presence of 3′ intra/inter primer homology was ruled out using IDT OligoAnalyzer tool (http://eu.idtdna.com/analyzer/Applications/OligoAnalyzer/) and the availability of the target sequence was evaluated by prediction of the cDNA secondary structure using Mfold web server[53]. Primer sequences are reported in Supplementary Table 1. The PCR reaction was performed with GoTaq qPCR Master Mix (Promega #A6002) and run in 7500 Fast Real-Time PCR System (Applied Biosystems), using following conditions: 95 °C 5 min; 45 cycles of 95 °C 15 s and 60 °C 45 s. The calculations were performed following $2^{-\Delta\Delta CT}$ method [CT(control)−CT(experiment)], where the control was calculated as the average CT value deriving from control samples. The normalization was performed using *TBP*. The statistical significance was calculated using the ΔCT values [CT(gene of interest)−CT(reference gene)] for each biological replicate in a group and applying Student's $t$-test[54]. Primer sequences are available in Supplementary Table 1.

**Fly lines and treatments.** Fly lines: yw, hs-Flp; l(2)gl$^4$/CyO; UAS-luc$^{KD}$—yw, hs-Flp; l(2)gl$^4$/CyO; UAS-NDUFV1$^{KD}$—w; l(2)gl$^4$/CyO; act:CD2:Gal4, UAS-GFP/TM6b—w; l(2)gl4, FRT40A/In(2LR)GlaBc; UAS-Ras$^{V12}$, UAS-luc$^{KD}$/TM6b—w; l(2)gl4, FRT40A/In(2LR)GlaBc; UAS-Ras$^{V12}$, UAS-NDUFV1$^{KD}$/TM6b—yw,

hs-flp, tub-Gal4, UAS-GFP; tub-Gal80, FRT40A. For experiments in Fig. 2f, larvae of the right genotypes were selected at $144 \pm 2$ h development, transferred in a 1.5 mL vial plugged with foam and immersed for 2 min in a water bath at 37 °C. After the heat-shock, larvae were immediately transferred onto fresh food and allowed to grow for additional 48 h before dissection. For experiments in Supplementary Fig. 5c and Supplementary Fig. 9e, larvae were heat-shocked at $48 \pm 4$ h development for 10 min in a water bath at 37 °C and allowed to grow for additional 72 h before dissection. All the experiments were carried out at 25 °C. In all the control experiments, the UAS-luc[KD] line was used as an irrelevant dsRNA. For validation of the VI[KD] construct, UAS-VI[KD] flies were crossed with act-Gal4 flies; the progeny was collected at the end of the larval life and processed as to obtain the total RNA for a qRT-PCR analysis.

**Drosophila disc isolation and volume calculation.** For experiments in Fig. 2f, $192 \pm 2$ h larvae displaying GFP+ cells were selected under a Nikon SMZ1000 fluorescence stereoscope. Collected larvae were dissected in cold PBS, tumorous imaginal wing discs were isolated and photographed. Major and minor axes were measured for each wing disc with ImageJ (NIH) and volumes were calculated approximating disc shape to a spheroid with depth = width. A minimum of 25 discs were analyzed for each replicate.

**Immunofluorescent staining.** For xenograft analysis, the samples were formalin fixed following standard protocols. Tissue sections (4 μm) were deparaffinised in xylene, rehydrated in absolute 2-propanol followed by citrate antigen retrieval (10 mM sodium citrate, pH 6) for 15 min at 95 °C and 20 min at RT. Blocking was performed with goat serum (Abcam #156046) for 10 min at RT and incubation with Alexa Fluor secondary antibodies (488-goat anti-mouse diluted 1:500 and 555-goat anti-rat diluted 1:350) for 40 min at RT. The following primary antibodies were used: rat anti-Endomucin (1:200, Santa Cruz #SC-65495) and mouse anti-SMA (1:750, Dako #M0851). Slides were mounted with Vectashield Antifade Mounting Medium with DAPI (Vector Laboratories, #H-1200). Vessel size was evaluated by measuring the longer diameter of 30 endomucin positive cells per tumor and avoiding areas of collective fibroblast infiltration. Fibroblasts (SMA+Endo−) and immature vessels (Endo+SMA−) were counted in five fields of view at ×20 magnification per tumor. Images were taken with Zeiss Axio Scope. Z1 scanner using ZEN software (Carl Zeiss Microscopy GmbH, Germany). For pimonidazole staining, 8 μm sections of snap-frozen samples were prepared and FITC-conjugated mouse monoclonal anti-pimonidazole antibody (1:400, Hypox-yprobe #Mab-4.3.11.3) was used following manufacturer's indications. In vitro immunofluorescence for anti-HIF-1α (1:350, Sigma-Aldrich #HPA001275) was performed using manufacturer's indications. Images for pimonidazole and HIF-1α staining were taken with an inverted Nikon Eclipse Ti-U epifluorescence microscope, using Metamorph software (Universal Imaging). For Drosophila experiments, tissues isolated from selected larvae were fixed and stained according to standard protocols. Primary antibodies: rabbit anti-cleaved caspase 3 (1:100, Cell Signaling #9961), mouse anti-MMP1 (1:50, DSHB, Iowa University), rabbit anti-Sima/HIF-1α[55], mouse anti-NimrodC1 (original antibody by Istvan Andó sent by Julia Cordero), rabbit polyclonal anti-phospho-histone H3 (Ser10) (1:100, Cell Signaling Technology #9701). Secondary antibodies: anti-mouse 555 Alexa Fluor (1:200) and anti-rabbit Cy5 DyLight (Jackson Laboratories, 1:500). Confocal images were processed as a whole with Adobe Photoshop. All the images shown represent a single confocal stack. Clone area, clone roundness and macrophage area were calculated by using ImageJ (NIH). For clone area (Fig. 2f) and roundness (Supplementary Fig. 5c), 20 random clones from separate discs were imaged. For cell density and caspase 3 (Supplementary Fig. 5b) quantification, marked cells present in four areas of $100 \times 100$ μm captured at different focal planes were counted for each tumor, for a total of 6 and 10 tumors, respectively.

**Flow cytometry.** Cell isolation was performed as previously described[56]. Briefly, xenograft samples (~50 mm³) were digested immediately after the sacrifice for 40 min at 37 °C with Liberase TL (Sigma #5401020001), Liberase TM (Sigma #5401135001) and DNaseI (Sigma #DN25) in HBSS and passed through a 100 μm strainer. Hypotonic lysis with Red Blood Cell Lysis Buffer (Sigma #11814389001) was performed and remaining cells were washed with MACS buffer (2 mM EDTA, 0.5% BSA in PBS), blocked using FcR Blocking Reagent (Miltenyi #130-092-575) and incubated with panels of pre-labeled antibodies. In parallel, spleen, lung and a control tumor tissue were digested together and stained for fluorescence minus one (FMO) reading which was considered while setting the gating strategy. Following panels were used: Panel 1 (for discrimination of non-cancer versus cancer cells, and for the analysis of the immune cell/fibroblast contribution in the tumor): anti-CD298-APC (clone LNH-94, Biolegend #341706), anti-CD45-APC780 (clone 30-F11, eBioscience #47-0451-80), and anti-CD31-PECy7 (clone 390, eBioscience #25-0311-82); Panel 2 (for analysis of the tumor macrophage, neutrophil, natural killer cell and dendritic cell contribution): anti-anti-CD45-PE (clone 30-F11, eBioscience #12-0451-82), anti-CD11b-ef450 (clone M1/70, eBioscience #48-0112-82), anti-F4/80-APC780 (clone BM8, eBioscience #47-4801-80), anti-Ly6G-APC (clone 1A8, BD Bioscience #560599), anti-CD11c-PECy7 (clone N418, Biolegend #117317), and anti-CD49b-FITC (clone 30-F11, Biolegend #108905). Panel 3 (for analysis of M1/M2 protumorigenic macrophages): anti-CD45-BV421 (Biolegend #103133),

anti-F4/80-APC780 (clone BM8, eBioscience #47-4801-80), anti-CD206-APC (clone C068C2, Biolegend #141707), anti-Arg1-PECy7 (clone A1exF5, eBioscience #25-3697-82), and anti-iNOS-PE (CXNPT, eBioscience #12-5920-82). Panel 4 (for analysis of monocyte differentiation): anti-CD45-BV421 (Biolegend #103133), anti-CD11b-PECy7 (clone M1/70, Biolegend #101215), anti-Ly6C-APC (Biolegend #128016), and anti-F4/80-APC780 (clone BM8, eBioscience #47-4801-80). All antibodies were used at 1:100 dilution, apart from the anti-CD45 which was diluted 1:300. Between 300,000 and 500,000 cells were stained. Dead cells were stained with DAPI. For Arg1 and iNOS intracellular staining Intracellular Fixation and Permeabilization Buffer Set (eBioscience #88-8824-00) was used by following manufacturer's indications, together with LIVE/DEAD Fixable Blue Dead Cell Stain (Invitrogen #L34962). The samples were run on LSRFortessa cell analyzer (BD Biosciences) and data was analyzed by BD FACSDIVA Software (BD Bioscience) and Flow Jo (Tree Star Inc.) software.

**Gel contraction assay.** To assess fibroblast activation toward a CAF phenotype, normal murine-derived breast fibroblasts[57] were seeded in 24-well dishes at 20,000 cells/well in 100 μL of 2:1 Rat tail collagen-I (Corning #354249) and Matrigel (Corning #356234), yielding a final collagen concentration of 4 mg mL−1 and a final Matrigel concentration of 2 mg mL−1. Conditioned media from a 48 h culture of 100,000 cancer cells (500 μL = 1/4 of the volume), was added and gel-detachment from the well edge was followed.

**Cytokine array.** $Rag1^{-/-}FVB/n$ Xenograft-derived cell cultures were generated by a 10-day cultivation of liberase-digested tissue in basal conditions. Supernatant (0.5 mL) was taken 2 days after medium renewal and analyzed with human Proteome Profiler Array kit (R&D Systems, ARY005B) following manufacturer's instructions.

**Statistical analysis.** GraphPad Prism version 7 (GraphPad Software Inc., San Diego, CA, USA) was used to perform statistical tests and create bar plots and graphs. Unless stated otherwise, a two-tailed unpaired Student's $t$-tests assuming equal variance were performed to compare averages. When the F-test to compare variances between the two groups was significant, the data were transformed ($y' = \log y$) prior to the $t$-test calculus. In the few cases where log transformation did not correct variances, a $t$-test for unequal variances was applied. For each experiment, at least three biological replicates were analyzed. In vitro analyses were repeated by at least two independent experiments. In vivo experiments were repeated at least twice for data presented in Fig. 1a, Fig. 3g, Fig. 5a–c, Fig. 6b, Fig. 6d, Supplementary Fig. 16a, and Fig. 17c. For each experiment, $p$-values (*$p < 0.05$, **$p < 0.01$, ***$p < 0.001$), $t$-values ($t$) and degrees of freedom (df) are indicated in figure legends, as well as the specification whether the log transformation or the $t$-test for unequal variances was performed. Moreover, where indicated, standard error of the mean (s.e.m.) is represented by the error bars. Survival curves were estimated using the Kaplan–Meier product-limit method and compared using a log-rank test (Mantel–Cox).

**Reporting summary.** Further information on experimental design is available in the Nature Research Reporting Summary linked to this article.

## Data availability
The authors declare that data supporting the findings of this study are available within the paper and its supplementary information files.

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

## Acknowledgements

This work was supported by EU H2020 Marie Curie project TRANSMIT GA 722605 and EU FP7 Marie Curie ITN-317433 MEET to A.M.P. and G. Gasparre; by Associazione Italiana Ricerca sul Cancro (AIRC) grant JANEUTICS-IG14242 and Italian Ministry of Health grant DISCO TRIP GR-2013-02356666 to G. Gasparre. GG. Girolimetti was supported by AIRC triennal fellowiship "Livia Perotti" and M.D.L. by triennial AIRC fellowship "Bruna Martelli". This work was also supported by The Francis Crick Institute which receives its core funding from Cancer Research UK (FC001112), the UK Medical Research Council (FC001112), and the Wellcome Trust (FC001112). We thank the Biomedical Research Facility, Experimental Histopathology and Flow cytometry units from The Francis Crick Institute (UK). We are also grateful to Sepideh Aminzadeh-Gohari, Felix Locker, Simone Di Giacomo, Patty Wai and Claudia Calabrese for technical help. We finally thank Christine M. Betts for English language editing and Pasquale Chieco for the help with statistical analyses.

## Author contributions

I.K. and L.B.A. created the NDUFS3$^{-/-}$ models and participated in the generation of inducible NDUFS3 knockout and TM-HIF-1α models. I.K. evaluated HIF-1α protein levels and HIF1-responsive gene expression. L.I., G.L. and R.V. performed biochemical experiments. I.K. and R.V. performed in vivo studies and histology analyses. M.D.L. participated in creation of the inducible NDUFS3 knockout model. G. Girolimetti, N.U.G., V.L.B., R.G.F., S.V., L.O. and M.B. provided technical help for in vivo studies.

M.C. performed electron microscopy analyses. M.R. is the pathologist who evaluated all histology data. I.K. performed flow cytometry experiments, for which L.O. and I.M. helped with data analysis. L.G. and A.C. performed vector transductions. M.V. created the vector containing TM-HIF-1α. M.S., S.F. and D.G. performed fly experiments. A.M.P. and G. Gasparre conceived the study. I.K., L.I., A.M.P. and G. Gasparre wrote the manuscript. L.O., I.M., B.K., R.V. and S.F. critically revised the manuscript.

## Additional information

**Competing interests:** The authors declare no competing interests.

