## [Peer Review File · Nature Communications]

Reviewers' Comments:

Reviewer #1:

Remarks to the Author:

In this manuscript, Kurelac et al report their findings of features of cells and tumors carrying inactivation of the mitochondrial complex I and of a possible therapeutic strategy to "induce indolence" in tumors.

Using zinc finger nuclease (ZFN) -mediated knockout of NDUFS3, they abolished complex I activity in two cell lines. They demonstrate that knockout of NDUFS3 decreases tumor growth and imparts features reminiscent of oncocytoma in xenograft models. They also utilize a *Drosophila* model system to show that lack of complex I activity is associated with tumor growth inhibition.

Mechanistically, the authors identify lack of HIF1a stabilization as a possible mechanism of tumor growth inhibition. Further, they report that HIF1a-independent tumor stromal changes including macrophage infiltration as potential mechanisms of escape from the tumor growth inhibition caused by complex I deficiency.

This report is interesting and highlights the role of oxidative phosphorylation as an important aspect of cancer metabolism. Multiple recent findings have conclusively demonstrated that there is more to cancer metabolism than the Warburg phenomenon alone. While the manuscript is generally well written, it would benefit from thorough copy editing to improve readability.

The experiments are generally well executed and appropriate controls are used in most experiments. While the two cell lines and xenografts derived from them are good model systems, the report would benefit tremendously if pharmacologic methods of inhibiting mitochondrial complex I were also used in parallel (at least for some crucial experiments) to show the relevance of their findings to potential therapeutics that might benefit patients.

This reviewer has several concerns that need to be addressed before the manuscript is further considered for publication.

1. The authors show a discrepancy in the effect of NDUFS3 knockout on cell proliferation in vitro and in vivo but fail to address the underlying mechanism (Fig. 1A & Suppl. 2H): 143B NDUFS3 WT and KO cells have similar proliferation rate in vitro, whereas HCT116 NDUFS3 knockout cells have reduced proliferation compared to WT cells. In addition, NDUFS3 KO cells have low Ki67-proliferation index.

2. In Fig. 1B, there is significant variability of Ki67 staining cells in the 143B WT and KO cells (N=15). The authors only show Ki67 staining for 3 of the HCT116 WT and KO tumors even though N=8. The authors need to explain why they left out the other 5 tumors.

3. Figure 1E-1F & Supp. Fig. 3B. The authors show that PPARGC1A, the master regulator of mitochondrial biogenesis, is increased in 143B KO cells and re-expression of NDUFS3 in 143B^{-/-} cells rescued complex I activity, mitochondrial ultrastructure and tumorigenic potential. How about HCT116 KO cells?

4. The authors convincingly show destabilization of HIF1a in complex I deficient cells in vitro and nominate this as a primary mechanism for the impaired tumorigenic potential of complex I deficient cells. However, this is not formally proven by genetic epistasis experiments. The authors should repeat the experiments reported in Fig 3G by including several additional controls including knockdown of HIF1a in the parental 143B and HCT cell lines, as well as introducing the HIF1a-TM mutant in the parental 143B and HCT cell lines.

5. The authors speculate that HIF1a is destabilized in complex I deficient cells because of increase in alpha-KG and subsequent activation of PHD enzymes. However, this is not supported by data provided. The authors should use cell permeable analogs of alpha-KG (at similar concentration to what they observe when complex I is knocked out) in their model systems and measure HIF1a protein levels, measure cell growth and analyze gene expression similar to Figure 3A, Supplementary Fig 2 and Figure 3B respectively. Furthermore, the authors need to rescue HIF1a degradation by knocking down or knocking out PHD enzymes in 143B^{-/-} and HCT^{-/-} cell lines.

6. The authors used different FCCP concentrations for the Seahorse experiment in Supp. Fig 2. They need to compare data using the same concentration in both the parental cell line and its knockout counterpart.

7. Fig. 2B. The authors should show the Ki67 staining of the tumors in dox-inducible NDUFS3

expressing 143B^{-/-} cells.

8. Supp. Fig. 2H. The y-axis should not be viability (%).

9. Fig. 3D. The authors claim that the rare hypoxic foci identified in 143B^{-/-} 209 xenografts were negative for HIF-1 α staining. However, the image shown and a lack of quantification of HIF1 α positive cells do not support their conclusion. They need to show hypoxia probe pimonidazole staining in additional tumors.

10. Data presented in Supp. Fig 4 is plotted in an unconventional manner. Authors need to plot -Dox and +Dox all in one graph (both mean values and spider plots would be useful if there is high variability between tumor growth rates across individual mice).

11. Fig. 4D. The authors claim that an increase in α -KG concentration and reduced oxygen consumption contributed to an increase in PHD activity. However, α -KG level is not statistically different in 143B WT and KO cells. In addition, the authors should determine the expression of PHD1, PHD2, and PHD3 in the WT and KO cells.

12. Fig. 5B. The authors should also show MT and CD31 staining for 143B WT and HCT116 WT cells.

13. Fig. 7B-D. The authors show a decrease in MIF abundance in 143B KO cells conditioned medium; they should use conditioned medium to perform in vitro migration assay to evaluate macrophage recruitment.

14. Fig. 7E. The authors use clodronate to deplete macrophages and demonstrate that targeting macrophages can potentiate the effect of complex I-targeting agents. To address clinical relevance, the authors should test the effect of a CSF1R inhibitor.

15. Data provided in Figures 5-7 showing the quantification of various tumor stromal components are inconsistently variable, not significant or barely significant. As it is widely known that the tumor microenvironment shows tremendous heterogeneity, these findings are not surprising. However, with the limited number of cell lines used in their study it is very difficult to draw any firm conclusion with any degree of reliability. The authors suggest that epithelial cell lines with Complex I deficiency might be more dependent on microenvironment than stromal. This statement cannot be made using a single epithelial and a single mesenchymal cell line and need to be removed or supported by sufficient number of cell lines. Overall, the data provided in figures 5-7 are weak and unconvincing. The authors should utilize additional cell lines to convincingly show the role of macrophages in growth of complex I-deficient tumors. Furthermore, they should repeat experiment in Figure 7E with the parental HCT and 143B cell lines in parallel to complex I knockout lines to show the relative dependence of these tumors to depletion of macrophages. Without such comparison, it is impossible to judge whether the effect of clodronate in Fig 7E-F is really selective for complex I-deficient tumors or a general non-specific behavior observed when macrophages are depleted in any tumor.

16. The authors should include high resolution and high magnification images for all the IHC data they provide including insets focused on few cells to demonstrate intensity of staining. It is nearly impossible to make any conclusions for readers based on the images provided.

17. The authors should list number of animals used for all in vivo experiments in the figure legends.

18. The authors chose mesenchymal (osteosarcoma 143B) and epithelial (colorectal cancer HCT116) cancer cells as model systems to generate NDUFS3 cells, but they didn't explain what drove this choice.

19. Even though the authors show a western blot of 2 WT clones and 2 knockout clones in Supp. Fig. 1D, It is unclear how many NDUFS3 knockout single clones were generated and how they were used in the experiments (a pool of multiple single clones or just used one single clone for the experiments). If only one single clone was used, it will be essential to show that other single clones also displayed the same phenotype given the potential heterogeneity at the single cell level.

20. As of the innate immune system characterization, the authors only analyzed macrophages (F4/80+Lys6G⁻), neutrophils (Lys6G+F4/80-331) and dendritic cells (CD11c+F4/80-332). Given that infiltration of NK cells was shown to be high in renal oncocytomas [Geissler K. et al, Oncoimmunology. 2015 Jan; 4(1): e985082], it would be important to more comprehensively profile the innate immune system in the WT and KO tumors.

21. The authors only analyze the proliferation phenotype of knockout cells. Given that He X. et al [PLoS One. 2013 Apr 22;8(4):e61677] showed that knockdown of GRIM-19 or NDUFS3 of the complex I increases cell migration and invasion, it might be important to test whether there is any difference in the migration and invasion for the NDUFS3 knockout and WT cells.

22. Regarding the role of HIF1a in the angiogenesis and macrophages recruitment, in the abstract the authors mention that complex I-deficient tumors survive by triggering non cell-autonomous mechanisms of angiogenesis, independent of HIF-1a. However, they provide no experimental evidence to that claim in the manuscript.

23. On page 23 the authors mention that macrophage recruitment in complex I-deficient tumors may be a consequence of HIF1-MIF axis inactivation. They should evaluate the expression of pro-angiogenic factors in complex I-deficient tumors and whether HIF1a can regulate them.

Reviewer #2:

Remarks to the Author:

The overall goal of this study is to investigate the potential of modulating respiratory complex I (CI) to reverse the aggressive phenotype of tumors into indolent oncytomas by causing oxidative phosphorylation defects. The authors use two models of xenograft tumors in which a specific enzyme in this pathway, NDUFS3, is depleted constitutively or conditionally, using Doxycycline inducible KO. They found that CI inhibition leads to a block in cancer cell proliferation in mesenchymal and epithelial tumors, partly through HIF1a deregulation and loss of function. Indeed, the authors show that restoring a non-degradable form of HIF1a compensate for CI defects in xenografts and in drosophila tumor models. The slow but progressing nature of CI-deficient tumors seem to rely on changes in their tumor microenvironment, and particularly fibroblasts and macrophages. The numbers of total macrophages which display an M2-like phenotype, as characterized by CD206 expression, is increased in CI deficient tumors, and expression of MIF, a HIF1-regulated pro-inflammatory cytokine, is reduced in these tumors. The authors show that limiting macrophage infiltration by clodronate liposomes in CI deficient tumors inhibits their ability to reinitiate growth. The authors suggest that dual approach targeting CI complex and immune cells could be advantageous to maintain aggressive tumors in an oncytoma-like, stable state.

General comments

This study addresses the potential of interfering with oxphos via CI complex targeting, to induce indolence in malignant tumors, and elaborates on the adaptive mechanisms responsible for escape routes in the context of CI inhibition that could mediate tumor regrowth. Inability of tumors to accommodate hypoxia, and alternative approaches changing the metabolic states of tumors represent attractive and novel therapeutic avenues, yet these strategies still lack mechanistic insights, which makes this study relevant. Altogether, the experiments are well performed and thorough, the figures are visually attractive, and the topic is novel. The results presented justify the authors' conclusions, but call for important clarifications and additional experimental insights, as pointed out in the comments below:

Major comments:

1- Recent reports (in which the authors have participated; Kurschner et al 2017) identified increase of glutathione as a new metabolic hallmark in renal oncytomas. Are GSH levels altered in tumors KO for NDUFS3 compared to CI-proficient xenografts? Does this relate to ROS levels?

2- The authors target NDUFS3 to induce an mtDNA-independent model of CI dysfunction. Their results in these cells shows that glycolysis is upregulated together with reductive carboxylation. While similar effects have been reported for metformin in cancer cells, it would be important to show similar effects on tumor growth in the context of metformin treatment of these tumors, and validate the effects they report on in vitro and in vivo differential effects in the two cell lines used.

The effect of metformin treatment on macrophage localization/activation should also be examined to anticipate similar adaptive mechanisms effects of treated tumors. Indeed, as mainly NDUFS3 KO is used in this study, an equivalent mode of CI inhibition or mtDNA mutations would complement the authors findings.

3- The biochemical characterization of NDUFS3 KO cells is largely performed in vitro, and as the authors point out, this does not take into account the selective pressure of an in vivo environment. Additional data on the bioenergetic features of these tumors should be included, either by analyzing the tumors themselves, or by putting the cells back in cell culture and comparing the changes associated with in vivo selection.

4- The timing of when tumors re-expressing NDUFS3 in the Dox+ experiment catch up with the control Dox- treated ones is unclear, and the variability of response (SuppFig4) in this system should be discussed and correlated with efficacy of NDUFS3 KD or differences in CI inhibition/metabolic changes.

5- It is somewhat intriguing that a mature and organized vasculature is a characteristic of the more aggressive CI competent tumors. What is the functionality of these vessels in the different constitutive or inducible NDUFS3KO tumors compared to the relevant controls?

6- A more complete analyses of immune cells should be performed, including Ly6C and Cd11b at the minimum to determine monocyte recruitment in parallel to macrophages, as it is a possibility that the metabolic differences in these tumors affect monocyte differentiation.

7- The use of CD206 as sole readout of M2-like macrophage phenotype is insufficient, as the spectrum of activation of these cells is complex and plastic. The authors should use additional markers and in any case, be mindful of overinterpretation based on a handful of markers.

8- As the authors infer that infiltration of macrophages into the tumor bulk mediates the compensatory phenotype in NDUFS3 KO tumors allowing to acquire a more malignant phenotype, it would be important to assess the intratumoral and peritumoral localization of CAFs and macrophages at d10, and determine whether these cells infiltrate the tumors bulk to favour the acquisition of adaptive mechanisms that may participate to revert tumor indolence, or if they are present within the tumor mass in the early onset of CI-deficient neoplasia.

9- The tumor growth curves presented in Fig 7E and Fig 1A are strikingly different, this is concerning- why did the authors decided to change from nude to Rag1-/- mice?

10- The clodronate liposome treatment is likely to affect CI-proficient tumors as well, this comparison should be included here, and also performed in the HCT model. Moreover, the depletion of monocyte/macrophages using this strategy, or others more efficient (eg CSF-1R inhibition) should be done in nude mice, and integrated when the increase in macrophage numbers is observed in CI deficient tumors (between d10 and d30).

Specific comments:

- The authors should justify the use of 143B and HCT116 cell lines for their study, is it only based on the aggressiveness of these tumor cell lines?

- While in the V1KD drosophila model, authors show that CC3 is not altered, comparison of cell death is not reported in the murine models, this should be corrected to ascertain that the inhibition on tumor growth are limited to proliferative defects and Ki67 or another proliferation read out should be performed in the drosophila model.

- The effect of TM-HIF mutant complementation in NDUFS3 KO cells on vasculature looks rather partial (Supp Fig 8D). Authors should perform quantitation and show additional representative images.
- The original scans should be included in Fig 7B, and quantitation should be provided.
- Typo line 160- 'cancer' not 'caner'
- Higher magnification images should be shown in Fig 3C to better observe the lack of HIF1 nuclear localization.
- Size bars are missing in multiple representative images
- Line 430-sentence is missing a word

Point by point responses to the Reviewers' comments.

We are grateful to the reviewers for their positive comments, and we agree that the introduction of pharmacological methods to inhibit CI would render the manuscript clinically relevant. Since our main aim was to understand what occurs when metformin is used in cancer therapy, due to its well ascertained role in inhibiting CI, we have now employed most of the revision effort in the attempt to translate our molecular findings into clinically relevant experiments *in vivo* with metformin, also in combination with macrophages inhibitors, as suggested by both reviewers. We have hence optimized the use of animals towards this end, with the hope that at least some of the limitations the reviewers have noted in our work and that have raised concern, may be surpassed now by these crucial experiments.

We would like to note that due to the introduction of the *in vivo* experiments on the combinatorial therapy, the title of the manuscript has now been slightly changed to better fit the novel content, into: "Inducing indolence in aggressive cancers by targeting mitochondrial Complex I is potentiated by blocking macrophage-mediated adaptive responses".

Reviewer #1 (Remarks to the Author):

In this manuscript, Kurelac et al report their findings of features of cells and tumors carrying inactivation of the mitochondrial complex I and of a possible therapeutic strategy to "induce indolence" in tumors. Using zinc finger nuclease (ZFN) -mediated knockout of NDUFS3, they abolished complex I activity in two cell lines. They demonstrate that knockout of NDUFS3 decreases tumor growth and imparts features reminiscent of oncocytoma in xenograft models. They also utilize a Drosophila model system to show that lack of complex I activity is associated with tumor growth inhibition. Mechanistically, the authors identify lack of HIF-1 α stabilization as a possible mechanism of tumor growth inhibition. Further, they report that HIF-1 α -independent tumor stromal changes including macrophage infiltration as potential mechanisms of escape from the tumor growth inhibition caused by complex I deficiency. This report is interesting and highlights the role of oxidative phosphorylation as an important aspect of cancer metabolism. Multiple recent findings have conclusively demonstrated that there is more to cancer metabolism than the Warburg phenomenon alone. While the manuscript is generally well written, it would benefit from thorough copy editing to improve readability. The experiments are generally well executed and appropriate controls are used in most experiments.

While the two cell lines and xenografts derived from them are good model systems, the report would benefit tremendously if pharmacologic methods of inhibiting mitochondrial complex I were also used in parallel (at least for some crucial experiments) to show the relevance of their findings to potential therapeutics that might benefit patients.

This reviewer has several concerns that need to be addressed before the manuscript is further considered for publication.

1. The authors show a discrepancy in the effect of NDUFS3 knockout on cell proliferation in vitro and in vivo but fail to address the underlying mechanism (Fig. 1A & Suppl. 2H): 143B NDUFS3 WT and KO cells have similar proliferation rate in vitro, whereas HCT116 NDUFS3 knockout cells have reduced proliferation compared to WT cells. In addition, NDUFS3 KO cells have low Ki67-proliferation index.

We reasoned that the focus of our work should not be to unravel the mechanisms underlying *in vitro* discrepancies. *In vitro* most selective pressure are lacking, the main of which being a shortage of nutrients, and cooperativity mechanisms among different types of cells within the tumor mass may not be investigated. Discrepancies between the *in vitro* and the *in vivo* situation, and among

different cells lines, which likely are differently adapted in terms of metabolism also as a consequence of diverse modifier genetic lesions, fell rapidly out of the scope of our work when we were certain that, despite growth differences *in vitro*, all models we used (both cells and *Drosophila*) behaved coherently *in vivo*. We hence turned our attention to what we considered to be more relevant for a translational research, i.e. the *in vivo* results. Understanding the cell proliferation data may generate confusion, we have now removed them from the Supplementary Figure 2 and replace them with what we consider more relevant, i.e. the invasive potential of cells, which is now reported in Figure 1B (please see also our response to comment 21).

2. In Fig. 1B, there is significant variability of Ki67 staining cells in the 143B WT and KO cells (N=15). The authors only show Ki67 staining for 3 of the HCT116 WT and KO tumors even though N=8. The authors need to explain why they left out the other 5 tumors.

As an intrinsic feature of studies in which one manages to notably reduce the tumor mass in a group of animals, the available material may be quite scarce, as xenografts remain very small. In our case, CI KO tumors are indeed quite small, and we come to a point when we need to balance out the number of animals used, which must be kept large enough to ensure statistical significance, but ethically minimal. As we show in the manuscript, the characterization of such masses has been thorough, and many different analyses and sampling have been carried out (cytometry on fresh tissue, RNA and protein analyses, electron microscopy and histology staining). We therefore proceeded with a cautious use of the samples available, reasoning that we would use the minimum available number of samples per analysis, and increase such number in case of ambiguous/not uniform results. With respect to KI67, we were able to observe a very coherent pattern and a clear-cut difference between the two groups (KO and WT), which was a confirmation of the macroscopic growth observed.

In the experiment for which more material was available we indeed counted the KI-67 in a much higher number of tumors, as depicted in the graph of Figure 2B.

3. Figure 1E-1F & Supp. Fig. 3B. The authors show that PPARGC1A, the master regulator of mitochondrial biogenesis, is increased in 143B KO cells and re-expression of NDUFS3 in 143B-/- cells rescued complex I activity, mitochondrial ultrastructure and tumorigenic potential. How about HCT116 KO cells?

We apologize with the reviewer for not specifying in the manuscript that HCT116 cells express undetectable levels of PPARGC1A, as previously reported (Bartoletti-Stella et al., Cell Death and Disease 2013). In this study, indeed, we detected expression at $ct > 32$ with 200ng of RNA used for cDNA synthesis, and therefore data could not be considered for the analysis. Being this an ancillary datum, merely as a confirmation of the clear-cut mitochondrial biogenesis we observed, we did not delve into the mechanisms underlying the mitochondrial mass increase occurring in HCT116, which may easily be due to the genes of the PPARGC family.

However, should this datum generate confusion rather than strengthening the observation of an ongoing oncocytic transformation when CI is abolished, we are willing to remove it from the paper, as it will not change the overall message. Again, here we only intended to state that at least in the model expressing PGC1a, mitochondrial biogenesis was activated via this mechanism.

4. The authors convincingly show destabilization of HIF-1 α in complex I deficient cells in vitro and nominate this as a primary mechanism for the impaired tumorigenic potential of complex I deficient cells. However, this is not formally proven by genetic epistasis experiments. The authors should repeat the experiments reported in Fig 3G by including several additional controls including knockdown of HIF-1 α in the parental 143B and HCT cell lines, as well as introducing the HIF-1 α -TM mutant in the parental 143B and HCT cell lines.

We thank the reviewer for this remark. We have managed to successfully obtain parental 143B cells expressing HIF-1 α -TM, to complete the experiments we have reported in the manuscript. These data, showing that there is no epistatic effect of the transcription factor *in vivo*, have now been included in the Results section (Supp. Fig. 8D).

On the other hand, with respect to the knock-down of *HIF-1A*, the finding that it slows down tumor progression has already been extensively shown in several cancer models (Schwab LP et al, 2012; Gillespie DL et al, Clin Cancer Res. 2007), also by authoritative groups such as Semenza's (Lee K et al, PNAS, 2009) and Simon's group (Shay JES et al, 2014, Carcinogenesis). We reasoned therefore that such experiment would not, in this context, add value to the message of the work, as we are not focusing on mechanisms generally contributing to progression of parental 143B and HCT116 cells: our objective was to elucidate mechanisms through which CI dysfunction causes reduction of tumorigenic potential. By supplementing CI-deficient cells with the non-degradable form of HIF-1 α we show that the lack of this transcription factor contributes to the antitumorigenic effect of targeting CI.

5. The authors speculate that HIF-1 α is destabilized in complex I deficient cells because of increase in alpha-KG and subsequent activation of PHD enzymes. However, this is not supported by data provided.

The authors should use cell permeable analogs of alpha-KG (at similar concentration to what they observe when complex I is knocked out) in their model systems and measure HIF-1 α protein levels, measure cell growth and analyze gene expression similar to Figure 3A, Supplementary Fig 2 and Figure 3B respectively. Furthermore, the authors need to rescue HIF-1 α degradation by knocking down or knocking out PHD enzymes in 143B $^{-/-}$ and HCT $^{-/-}$ cell lines.

In order to accomplish the reviewer's request, we have searched for α -KG analogs commercially available and obtained the Octyl-2KG, which we tested on our cell models to verify it decreases HIF-1 α stabilization in the millimolar concentration range. Unfortunately, this compound merely contributed not to increase HIF-1 α stabilization in hypoxia, in agreement with what has been previously reported by other groups (Hou P, et al., PLoS ONE 2014). We excluded the dimethyl-2-ketoglutarate, since on the same paper Hou et al show this is indeed a HIF-1 α stabilizer. Analogues previously used in the literature and shown to work, such as 3-trifluoromethylbenzyl α -KG ester, are not unfortunately commercially available. We apologize for not being able to carry out this requested experiment.

With respect to the suggestion of depleting PHD enzymes, we thought it more feasible to inhibit them pharmacologically and specifically: this is a long proven efficient method to obtain the rescue of HIF-1 α degradation, which is what the reviewer is asking, and much less cumbersome than for instance obtaining genetically manipulated models. We believe that the experiments we have performed and the data we have shown in Fig. 4B and Fig. 4C unequivocally prove the point that it is a higher activity of PHDs that leads to HIF-1 α destabilization, especially in the light of the data on the hydroxylated form of HIF-1 α we also provide.

In detail, we showed a pharmacological inhibition of PHD activity at 5-hour exposure to hypoxia (1% O₂) by DMOG, which results in the rescue of HIF-1 α stabilization in 143B $^{-/-}$ and HCT $^{-/-}$ cells. Moreover, by MG132 treatment at the same hypoxic conditions, we show specific accumulation of the hydroxylated form of HIF-1 α when proteasome is blocked, indicating that PHDs hydroxylation activity in KO cells is active despite the 5-hour exposure to hypoxia.

6. The authors used different FCCP concentrations for the Seahorse experiment in Supp. Fig 2. They need to compare data using the same concentration in both the parental cell line and its knockout counterpart.

Together with cell seeding, the use of an appropriate FCCP concentration is a critical aspect of Seahorse experimental procedure. Indeed, the manufacturer protocol highlights the importance of FCCP titration for the optimization of a microrespirometry assay of different cell lines. Indeed, cell types vary in their response to FCCP and doses of FCCP too high may result in reduced OCR responses. Hence, the manufacturer suggests to determine empirically the FCCP dose that generates the maximal OCR (please refer to the Basic Procedure protocol available on Seahorse web pages and to Divakaruni AS, et al., Analysis and interpretation of microplate-based oxygen consumption and pH data. *Methods Enzymol.* 2014;547:309-54). Following this suggestion, we always perform a preliminary experiment to determine the optimal dose of FCCP using growing concentrations of uncoupler. The FCCP concentration range spans from 0.01 μ M to 2 μ M. For the subsequent experiments, we always use the lowest concentration of FCCP that allows to stimulate OCR to its maximal values. This concentration value may depend on cell type or the presence of mutations in genes encoding for respiratory complexes subunits. In our experience, mutations affecting respiratory complexes activity impact on FCCP tolerance and cells carrying these mutations require lower concentrations of FCCP compared to wild type cells.

7. Fig. 2B. The authors should show the Ki67 staining of the tumors in dox-inducible NDUFS3 expressing 143B^{-/-} cells.

We apologize with the reviewer for overlooking this. We have now reported the KI-67 staining data in tumors in which NDUFS3 knock-out was induced by doxycycline. We have included these data in the manuscript as Fig. 2B and modified the text accordingly.

8. Supp. Fig. 2H. The y-axis should not be viability (%).

This panel was removed from the Figure (see response to comment 1).

9. Fig. 3D. The authors claim that the rare hypoxic foci identified in 143B^{-/-} xenografts were negative for HIF-1 α staining. However, the image shown and a lack of quantification of HIF1 α positive cells do not support their conclusion. They need to show hypoxia probe pimonidazole staining in additional tumors.

As stated in the manuscript, most of the CI-deficient tumors are normoxic. Hypoxic foci were identified only in two CI-deficient tumors, both of which were left to grow for over 60 days. We have shown the pimonidazole staining of one of these tumors in Fig. 3D and of the other in Supp. Fig. 7C (far right panel). The latter one showed completely negative staining for HIF-1 α , whereas the CI-deficient tumor presented in Fig. 3D showed 26% of HIF-1 α positive nuclei versus 82.6% found in equivalently hypoxic areas of the WT tumors. However, this quantification is possible only for this one sample (n=1), not allowing therefore any statistical analysis. We have decided to show the image anyway, as it supports the results of experiments performed *in vitro* in the controlled hypoxic conditions (Fig. 3E, Supp. Fig. 7D). Even if the quantification of HIF-1 α positive nuclei is lacking due to insufficient number of cases with hypoxic foci, we hope that the images with better resolution we have now provided in Fig. 3D, may remain a part of the paper and contribute to deliver the message that CI dysfunction causes defects in HIF-1 α stabilization.

10. Data presented in Supp. Fig 4 is plotted in an unconventional manner. Authors need to plot -Dox and +Dox all in one graph (both mean values and spider plots would be useful if there is high variability between tumor growth rates across individual mice).

In this experiment, the animals were sacrificed once they reached the end point (10% of body mass), which for the untreated mice started at day 14, continued on day 18, and at day 20 there was

only 1 animal left in the control group. Thus, the timing until which the growth curve with at least $n=3$ is feasible is day 20, when we may not appreciate the difference between the Dox- and Dox+ group. This is due to the fact that the half-life for CI is 9 days (Supp. Fig. 4B), thus the actual CI knock-out is obtained from day 10 onwards. Indeed, for most animals, it is after 10 days of treatment when the growth between the two groups starts to diverge, as shown by single +Dox plots in Supp. Fig 5. This information is lost if we plot the data as mean values.

Since this experiment was set up to generate a survival curve, we showed single graphs in Supp. Fig. 5 only with the aim to reveal the raw data. If the reviewer believes it is confusing, we may remove the Supp. Fig. 5 completely, but in our opinion it does provide informative data. In particular, it shows that the variability is relatively low, as only 1 out of 7 Dox+ mice did not respond to the induction of CI knock-out.

11. Fig. 4D. The authors claim that an increase in α -KG concentration and reduced oxygen consumption contributed to an increase in PHD activity. However, α -KG level is not statistically different in 143B WT and KO cells. In addition, the authors should determine the expression of PHD1, PHD2, and PHD3 in the WT and KO cells.

The difference in α -KG in 143B was not statistically different, since from 5 samples we originally sent for metabolomics analysis, α -KG was observed only in 2, likely due to the liability of this metabolite during samples preparation. In order to strengthen this datum, following the reviewer's remark, we have now performed further analyses and provided a graph with $n=5$, where the statistical difference may be appreciated (Fig. 4D). We have also assessed the expression of PHD1/2 protein levels, as the reviewer requested, and found that these enzymes were similarly expressed in NDUFS3 WT and KO cells, confirming that the increased HIF-1 α destabilization, or HIF-1 α hydroxylation, is not due to an increase in the amount of hydroxylating enzymes, but to their activity. We also tried to assess PHD3 levels by western blot, but the available antibody (NOVUS Biologicals #NB100-139) was sub-optimal. The data regarding PHD1/2 levels have now been reported in Supp. Fig. 9C.

12. Fig. 5B. The authors should also show MT and CD31 staining for 143B WT and HCT116 WT cells.

The CD31 staining in WT samples that the reviewer is here referring to is likely the one shown in Supp. Fig. 10A. The MT data have now been added as Fig. 5C, together with SMA staining to underline that CI-deficient tumors present with prominent stromal component, which in CI-competent tumors is mainly located on the growing tumor front. The data on the m.3571insC model of CI-deficiency have been moved to Supp. Fig. 10. The manuscript text, Fig. 5, Supp. Fig. 10 and figure legends were modified accordingly.

13. Fig. 7B-D. The authors show a decrease in MIF abundance in 143B KO cells conditioned medium; they should use conditioned medium to perform in vitro migration assay to evaluate macrophage recruitment.

We have attempted to address this question by performing a chemotaxis experiment in which RAW.264.7 and U937 cells were seeded in low serum media, whereas the conditioned media of either WT or KO tumor-derived cells was provided as the chemoattractant in the lower chamber of the Incucyte trans-well assay. We were not able to observe any signs of monocyte/macrophage migration in any of the experimental settings we tested, including coating the wells with collagen (Revision Figure 1). We have also tried to compare the migration of MACS-extracted F4/80 cells from CI-competent and deficient tumor masses, but again, we did not observe any signs of trans-well migration by using Incucyte chemotaxis setting. Thus, it proved to be difficult to reconstruct *in*

in vitro the phenomenon of higher infiltration of macrophages we clearly observe in tumor masses. This may be due to the fact that monocyte cell lines do not cope well in DMEM, which we are restricted to use in order to obtain conditioned media from 143B and HCT cells.

Thus, we removed the implication on the role of MIF in macrophage migration from the manuscript and restricted our observations to the correlation between MIF protein levels, macrophage abundance and HIF1 activity (lines 327-334 and Supp. Fig. 13). Furthermore, we acknowledge the possibility that MIF is not the only factor contributing to macrophage abundance in KO tumors (line 444-446), since, as suggested by the other reviewer, we now demonstrate a greater differentiation of monocytes to Ly6C⁺ macrophages in CI-deficient tumors (Fig. 6C, Supp. Fig. 11C).

Regardless of the mechanism linking macrophage abundance and CI-deficient cancer cells, we hope the reviewer agrees, we have proven the clinical importance of our finding, by demonstrating that targeting macrophage recruitment improves the effects of CI targeting.

Revision Figure 1. Incucyte chemotaxis images displaying cells lying on the top of the plate in orange/yellow and those who underwent trans-well migration in red. The images were taken 24-hours after addition of the conditioned media. No migration was observed in any setting involving monocyte/macrophage cells even after 72 hours of culture (data not shown).

14. Fig. 7E. The authors use clodronate to deplete macrophages and demonstrate that targeting macrophages can potentiate the effect of complex I-targeting agents. To address clinical relevance, the authors should test the effect of a CSF1R inhibitor.

We understand the reviewer's concern that the best way to prove the clinical relevance of our findings is to test the effects of a combined therapy, by using CI targeting agents and macrophage inhibitors *in vivo*. To this aim, we treated CI-competent xenografts with metformin and PLX-3397 (a CSF1R inhibitor also suggested by the other reviewer). We confirmed that the combination of metformin with PLX-3397 was the most efficacious therapy that dramatically decreased tumor growth *in vivo*, increasing the efficacy of using a single drug. These new *in vivo* experiments have now been added to the paper in the Results section and data are shown in Figure 7D. The abstract and the Discussion have now been modified accordingly.

15. Data provided in Figures 5-7 showing the quantification of various tumor stromal components are inconsistently variable, not significant or barely significant. As it is widely known that the tumor microenvironment shows tremendous heterogeneity, these findings are not surprising. However, with the limited number of cell lines used in their study it is very difficult to draw any firm conclusion with any degree of reliability. The authors suggest that epithelial cell lines with Complex I deficiency might be more dependent on microenvironment than stromal. This statement cannot be made using a single epithelial and a single mesenchymal cell line and need to be

removed or supported by sufficient number of cell lines. Overall, the data provided in figures 5-7 are weak and unconvincing. The authors should utilize additional cell lines to convincingly show the role of macrophages in growth of complex I-deficient tumors.

It is known that the data on microenvironment involvement in tumors are often heterogeneous, and we thank the reviewer for highlighting this point. In this context, we would like to remark that the higher contribution of macrophages in CI-deficient masses was always very consistent in our experiments, and we have shown this in three different models of CI-deficiency; (i) in two different cell lines with a knock-out for the nuclear subunit *NDUFS3*; (ii) in cells carrying the homoplasmic CI-disruptive m.3571insC mtDNA mutation in *MT-ND1*; and (iii) in *Drosophila* by knocking down the *NDUFV1* orthologue. In particular, in the *NDUFS3* KO models we observe higher contribution of macrophages both at days 10 and 30, and this was evaluated by two independent methods, namely cytometry and histology analyses. Histology analysis particularly showed a more clear-cut difference, as shown in Fig. 6D which underlies the difference in tumor infiltrating macrophages at day 10. In cytometry, the higher contribution of macrophages is more evident from the dot-plots, which have now been added to Fig. 6B. Regarding the cytometry data, it is important to note that CI-WT tumors, due to their aggressive nature and fast proliferation, harbor high number of necrotic cells (see Fig. 1D), which are excluded from the final analysis, meaning that the contribution of any stromal cell type in the WT tumor mass will be over-estimated. In addition, since CI-deficient masses harbor substantial number of stromal cells (see Fig. 5D), the percentage of a single stromal cell type among all live cells is under-estimated in CI-deficient tumors. Indeed, if we calculate the macrophage:cancer cell ratio, the difference between CI-competent and deficient tumors is even more evident. This information has also now been included in the manuscript (Fig. 6B). Finally, we have now included data showing higher abundance of macrophages in metformin treated tumors (Supp. Fig. 15D).

Thus, we believe that the association between macrophage recruitment and CI-deficient tumors is a quite robust datum. Although we may not agree with the reviewer that the data we provide on the contribution of macrophages in CI-deficient tumors are unconvincing and weak, we agree that the statement on the potential difference in dependence on the stroma between mesenchymal and epithelial cells may be speculative and we have now removed that sentence from the text.

Furthermore, they should repeat experiment in Figure 7E with the parental HCT and 143B cell lines in parallel to complex I knockout lines to show the relative dependence of these tumors to depletion of macrophages. Without such comparison, it is impossible to judge whether the effect of clodronate in Fig 7E-F is really selective for complex I-deficient tumors or a general non-specific behavior observed when macrophages are depleted in any tumor.

We apologize with the reviewer for not providing these experiments earlier, although in our frame of mind we did expect clodronate to have an effect on CI-competent tumors as well, since this is evident from the literature, as we had pointed out in the discussion (“Of note, CI inhibitors such as metformin and microenvironment-targeting agents, such as clodronate, both exhibit cytostatic effects on cancer cells.”). TAMs often accumulate in hypoxic tumor regions, where they contribute to tumor angiogenesis and invasiveness. Indeed, our control models do display hypoxia, as well as macrophage accumulation on the growing front (Fig. 6D), indicating that the macrophage function is in general crucial for tumor progression in solid tumors. We would therefore like to underline that by no means we intended to state that clodronate is selective for CI-deficient tumors. Nonetheless, we agree that parental cell lines treated with clodronate are needed to complete our data. We have now performed an experiment in which HCT CI-competent and CI-deficient cancers were both treated with clodronate and we do observe an anti-tumorigenic effect of clodronate in control tumors as well, as expected. The effect of clodronate on CI-deficient cells is nonetheless more

striking. Whereas clodronate treatment of CI-competent masses still allowed formation of frank tumors, in histology similar to their untreated controls, HCT^{-/-} xenografts treated with clodronate, even after 45 days, still formed masses not larger than 10mm³, which upon excision revealed to be completely white, indicating no blood perfusion, and on histology showed extensive necrosis, which is normally absent from CI-deficient masses (Fig. 7B and Supp. Fig. 17A-B). These data have now been integrated in the manuscript, and confirm that CI-deficient tumors are more sensitive to macrophage depletion.

16. The authors should include high resolution and high magnification images for all the IHC data they provide including insets focused on few cells to demonstrate intensity of staining. It is nearly impossible to make any conclusions for readers based on the images provided.

We apologize with the reviewer for this. We have now provided what requested. However, hoping that the reviewer agrees, in certain cases we preferred to maintain a lower magnification, in order to provide a general view of tumor histology.

17. The authors should list number of animals used for all in vivo experiments in the figure legends.

This information is available in the figure legends.

18. The authors chose mesenchymal (osteosarcoma 143B) and epithelial (colorectal cancer HCT116) cancer cells as model systems to generate NDUFS3 cells, but they didn't explain what drove this choice.

We believe that metabolic anti-cancer approaches that target hypoxic adaptation, such as CI inhibition, may potentially have an effect on any solid tumor. Thus, the aim in this study was to identify phenotypes developed upon CI KO, which are generalizable. This is why we chose cell lines deriving from different origin (mesenchymal *versus* epithelial). Moreover, the aggressive behavior of these cells is well known, as well as their growth kinetics *in vivo*. A sentence has now been added to clarify the point. An additional technical limitation only partially drove the choice, which had to do with the ploidy of the cell lines, as we left out cells that harbored more than two copies of the *NDUFS3* gene, in order not to incur in technical difficulties with the gene editing procedure.

19. Even though the authors show a western blot of 2 WT clones and 2 knockout clones in Supp. Fig. 1D, It is unclear how many NDUFS3 knockout single clones were generated and how they were used in the experiments (a pool of multiple single clones or just used one single clone for the experiments). If only one single clone was used, it will be essential to show that other single clones also displayed the same phenotype given the potential heterogeneity at the single cell level.

We tend not to use single clones after selection for our experiments since performing *in vivo* experiments would require a larger and unnecessary number of animals to take into account the potential differences in cells behavior. We therefore generated pools of clones and used them for all experiments. We now specify within the Method section the number of clones to constitute each pool. Since HCT could less bear the genetic KO of the *NDUFS3*, we generally obtained fewer clones to start with, which explains why fewer were used to constitute the pool. We would also like to assure the reviewer that the single clones were all tested for KO and for CI assembly and function before being pooled.

20. As of the innate immune system characterization, the authors only analyzed macrophages (F4/80+Lys6G-), neutrophils (Lys6G+F4/80-) and dendritic cells (CD11c+F4/80-). Given that

infiltration of NK cells was shown to be high in renal oncocytomas [Geissler K. et al, Oncoimmunology. 2015 Jan; 4(1): e985082], it would be important to more comprehensively profile the innate immune system in the WT and KO tumors.

We thank the reviewer for this suggestion. We have now analyzed mouse pan-NK cell marker CD49b in our models and integrated the data in Figure 6A. If NK cell contribution is calculated as % of all live cells, there seems to be a tendency for higher NK recruitment in CI-deficient tumors (Revision Figure 2), but their contribution to the whole mass is lower than 0.5%. We do not observe a significant difference in the percentage of NK cells among CD45 population. Thus, the biological significance of potential anti-tumorigenic effect of NK cells in KO tumors remains uncertain. Since we do not discover significant differences in NK cell recruitment, and since our aim is to identify microenvironment population which contributes to promote the growth of KO tumors, we do not provide further focus on NK cells. Nevertheless, we included the NK data to now offer a more complete innate immunity profile, which was furthermore substantiated by a more elaborate analysis of monocytes by CD11b and Ly6C markers (Fig. 6A and Fig. 6C, Supp. Fig. 12). The panels and gating strategy for this additional cytometry analyses were modified accordingly in Methods and Supplementary data.

[Redacted]

21. The authors only analyze the proliferation phenotype of knockout cells. Given that He X. et al [PLoS One. 2013 Apr 22;8(4):e61677] showed that knockdown of GRIM-19 or NDUFS3 of the complex I increases cell migration and invasion, it might be important to test whether there is any difference in the migration and invasion for the NDUFS3 knockout and WT cells.

Following the reviewer's suggestion, we tested the invasion capacity of our cell models in 3D revealing that CI KO cells are less invasive compared to their CI-competent counterpart. These data have now been added in Fig. 1B. It is important to note that we used xenograft-derived cultures, since these represent more appropriately the actual *in vivo* conditions than the original *in vitro* cell lines, which may explain why we observe a different result from what was published by He X. et al.

22. Regarding the role of HIF-1 α in the angiogenesis and macrophages recruitment, in the abstract the authors mention that complex I-deficient tumors survive by triggering non cell-autonomous mechanisms of angiogenesis, independent of HIF-1 α . However, they provide no experimental evidence to that claim in the manuscript.

We are sorry the sentence in the Abstract looked like an overstatement. We here intended that necessarily, due to the proven cells' inability to stabilize HIF-1 α , the angiogenesis we observe in CI-deficient tumors most likely ought to be promoted by HIF-1 α independent mechanisms mediated by the stroma. This rationale is based on the data that the lumen-bearing, pericyte positive vessels

are almost exclusively found in the stromal component of CI-deficient tumors (Fig. 5A-B, Supp. Fig. 10E). We have now included additional data showing that the number of F4/80 cells is decreased in CI-deficient tumors in which HIF-1 α activity is complemented by HIF-1 α -TM (Supp. Fig. 13D), suggesting that, when cancer cell mediated HIF-1 α signaling is present, the requirement for macrophage pro-tumorigenic activity is reduced. Moreover, we added data on Endomucin and SMA staining of clodronate or vehicle-treated CI-deficient tumors in Rag mice, which show no difference in the overall number of vessels in the tumor between the two experimental groups, but do reveal a specific reduction of SMA-negative vessels appearing in the tumor component of clodronate treated masses (Supp. Fig 17D). In tumors grown in nude mice, clodronate-treated CI-deficient masses are too small and completely necrotic, preventing us from drawing conclusions on angiogenesis. Nevertheless, the preliminary data from Rag experiment suggest that the angiogenesis observed in CI-deficient masses is most likely, in part, a result of macrophage activity.

As we understand that the reviewer is expecting here a formal proof that the angiogenesis is stroma-driven, and that the reduction of macrophages we observe when we re-instate HIF-1 α activity in KO tumors and the lower number of SMA- vessels in clodronate treated KO masses may not be considered sufficient evidence, we have now changed the Abstract sentence to avoid overstatements as follows: “We demonstrate that CI-deficient tumors survive and carry out angiogenesis, despite their inability to stabilize HIF-1 α ”. We also modified the sentence in the Results section concerning this issue into a simple working hypothesis that has driven our subsequent experiments: “we hypothesize that the stroma may be involved in compensatory mechanisms triggered to overcome the inability of CI-deficient tumors to activate HIF-1 α -mediated angiogenesis”.

23. On page 23 the authors mention that macrophage recruitment in complex I-deficient tumors may be a consequence of HIF1-MIF axis inactivation. They should evaluate the expression of pro-angiogenic factors in complex I-deficient tumors and whether HIF-1 α can regulate them.

[Redacted]

Reviewer #2 (Remarks to the Author):

The overall goal of this study is to investigate the potential of modulating respiratory complex I (CI) to reverse the aggressive phenotype of tumors into indolent oncocytomas by causing oxidative phosphorylation defects. The authors use two models of xenograft tumors in which a specific enzyme in this pathway, NDUFS3, is depleted constitutively or conditionally, using Doxycycline inducible KO. They found that CI inhibition leads to a block in cancer cell proliferation in mesenchymal and epithelial tumors, partly through HIF-1 α deregulation and loss of function. Indeed, the authors show that restoring a non-degradable form of HIF-1 α compensate for CI defects in xenografts and in drosophila tumor models. The slow but progressing nature of CI-deficient tumors seem to rely on changes in their tumor microenvironment, and particularly fibroblasts and macrophages. The numbers of total macrophages which display an M2-like phenotype, as characterized by CD206 expression, is increased in CI deficient tumors, and expression of MIF, a HIF1-regulated pro-inflammatory cytokine, is reduced in these tumors. The authors show that limiting macrophage infiltration by clodronate liposomes in CI deficient tumors inhibits their ability to reinitiate growth. The authors suggest that dual approach targeting CI complex and immune cells could be advantageous to maintain aggressive tumors in an oncocytoma-like, stable state.

General comments

This study addresses the potential of interfering with oxphos via CI complex targeting, to induce indolence in malignant tumors, and elaborates on the adaptive mechanisms responsible for escape routes in the context of CI inhibition that could mediate tumor regrowth. Inability of tumors to accommodate hypoxia, and alternative approaches changing the metabolic states of tumors represent attractive and novel therapeutic avenues, yet these strategies still lack mechanistic insights, which makes this study relevant. Altogether, the experiments are well performed and thorough, the figures are visually attractive, and the topic is novel. The results presented justify the authors' conclusions, but call for important clarifications and additional experimental insights, as pointed out in the comments below:

Major comments:

1- Recent reports (in which the authors have participated; Kurschner et al 2017) identified increase of glutathione as a new metabolic hallmark in renal oncocytomas. Are GSH levels altered in tumors KO for NDUFS3 compared to CI-proficient xenografts? Does this relate to ROS levels?

We have analyzed the GSH and GSSG levels in cells cultured in vitro and no significant difference was observed, neither in total glutathione, nor in GSH, GSSH or GSH/GSSH. These data have now been presented in Supp. Fig. 9B. The increase in glutathione observed in renal oncocytomas may be due to the restrictive nutrient availability in *in vivo* conditions, or possibly due to the contribution of

the stromal cell populations to the net concentration of the metabolite. As shown in Supp. Fig. 9A, we did not observe a significant difference in ROS levels between CI-competent and deficient cells. A tendency of CI-deficient cells to produce less ROS may be appreciated, most likely due to the fact that the complete lack of the complex abolishes also the ROS producing site.

2- The authors target NDUFS3 to induce an mtDNA-independent model of CI dysfunction. Their results in these cells shows that glycolysis is upregulated together with reductive carboxylation. While similar effects have been reported for metformin in cancer cells, it would be important to show similar effects on tumor growth in the context of metformin treatment of these tumors, and validate the effects they report on in vitro and in vivo differential effects in the two cell lines used. The effect of metformin treatment on macrophage localization/activation should also be examined to anticipate similar adaptive mechanisms effects of treated tumors. Indeed, as mainly NDUFS3 KO is used in this study, an equivalent mode of CI inhibition or mtDNA mutations would complement the authors findings.

We agree with the reviewer that it is important to show the action of metformin on tumor growth in the context of the tumors we use in the paper. To this aim, we have treated xenografts generated by our parental CI-competent cell lines with metformin, and shown that indeed the drug contributes to a tumor growth decrease. These data are now implemented in the manuscript in Supp. Fig. 15, together with the data recapitulating the macrophages phenotype shared by both metformin-treated and CI-KO xenografts. Additionally, we would like to draw the reviewer's attention to Supp. Fig. 15A, where we show that macrophage recruitment is higher in OS-93 tumors where CI deficiency is caused by a homoplasmic mitochondrial DNA m.3571insC mutation, when compared to their CI-competent OS-83 counterparts. Furthermore, an analogous phenotype is observed in the *Drosophila* model in which *NDUFV1* orthologue has been knocked down, where macrophage recruitment is again associated with the KO human cells. We have now also integrated data showing a tendency of higher F4/80+ abundance in metformin treated tumors as well (Supp. Fig 15D), and demonstrated that targeting this population by combining macrophage inhibitors with metformin increases the efficacy of a single drug treatment (Fig. 7D). We hope this helps convincing the reviewer that the macrophage recruitment in complex I-deficient tumors is a general phenomenon.

3- The biochemical characterization of NDUFS3 KO cells is largely performed in vitro, and as the authors point out, this does not take into account the selective pressure of an in vivo environment. Additional data on the bioenergetic features of these tumors should be included, either by analyzing the tumors themselves, or by putting the cells back in cell culture and comparing the changes associated with in vivo selection.

Indeed, we agree with the reviewer about the limitation of studying *in vitro* cells. We initially thought we should not focus on the metabolic reprogramming of CI-deficient cancers in order not to be redundant with what has been already ascertained authoritatively in the literature, i.e. that cancers gain a reductive carboxylation phenotype. Nonetheless, we agree that at least some biochemical characterization of cells re-cultured from xenografts ought to be performed to support the *in vitro* finding, keeping in mind that, as we previously showed, re-cultured CI null cells maintain the same bioenergetics profile as cells used for injection *in vivo* (Gasparre G., et al., Cancer Res, 2011). We hence validated the metabolic reprogramming of our *in vitro* cell models in re-cultured cells, and to this aim measured the lactate/pyruvate and aKG/citrate ratios, respectively indicative of glycolytic rate and reductive carboxylation, which showed analogue values to what is observed in the pre-*in vivo* cell lines. These data have now been integrated in Supp. Fig. 3A.

4- The timing of when tumors re-expressing NDUFS3 in the Dox+ experiment catch up with the control Dox- treated ones is unclear, and the variability of response (SuppFig4) in this system

should be discussed and correlated with efficacy of NDUFS3 KD or differences in CI inhibition/metabolic changes.

We thank the reviewer for raising this point. In Dox+ group one mouse reached end point at day 22 (mouse 7, see Supp.Fig.5) and two at day 30 (mouse 3 and 5). Day 30 was also when the last Dox-animal reached the end point so the experiment was ended 2 days after. We have now modified the Kaplan-Meier plot in order to make the results clearer. In our view, the response was not as variable as it might seem, since only 1 out of 7 mice did not respond at all to the induction of NDUFS3 knock out. Moreover, the homogeneous negative staining for NDUFS3 and NDUFS4 in Dox-treated tumors indicated the knock-down was efficient up to a functional KO, and no leakage of the promoter was observed (consistently with what we observed during *in vitro* characterization of the model). We do acknowledge there is a delay in response to the induction of NDUFS3 knock-out, but this is due to the fact that the half-life of Complex I is approximately 10 days. Indeed, consistent stalling of tumor progression is particularly evident between day 15 and day 30 *post* treatment.

5- It is somewhat intriguing that a mature and organized vasculature is a characteristic of the more aggressive CI competent tumors. What is the functionality of these vessels in the different constitutive or inducible NDUFS3KO tumors compared to the relevant controls?

It has to be acknowledged that aggressive tumors usually present with abnormalized vasculature. Indeed, SMA-negative vessels are also observed in NDUFS3 WT tumors (Fig. 5A), albeit at a much lesser amount than in NDUFS3 KO masses, which is the phenomenon relevant for the message of the study. Different models (m.3571insC OS-93 and inducible NDUFS3 KO tumors) both show the increase in abnormalized vasculature if compared to their controls. These data are now added to the manuscript (Supp. Fig. 10).

6- A more complete analyses of immune cells should be performed, including Ly6C and Cd11b at the minimum to determine monocyte recruitment in parallel to macrophages, as it is a possibility that the metabolic differences in these tumors affect monocyte differentiation.

We thank the reviewer for this comment, since by following his/her suggestion we have identified that, indeed, CI-deficient tumors present with higher monocyte differentiation. In particular, we now show in Fig. 6C and Supp. Fig. 12 that WT and KO tumors have an equal number of undifferentiated monocytes (CD11b+Ly6C+), but NDUFS3 KO masses have more CD11b+Ly6C-cells. Analogously, the number of F4/80+Ly6C+ is comparable between the two experimental groups, but the contribution of F4/80+Ly6C- macrophages is higher in KO tumors, all together indicating CI-deficient cancer cells promote monocyte differentiation. Interestingly, we observe that this phenomenon is also true for metformin treated masses (Supp. Fig. 15C-D). The text of the manuscript has been modified accordingly (line 317-323), as well as the Methods and Supplementary data explaining the gating strategies applied.

7- The use of CD206 as sole readout of M2-like macrophage phenotype is insufficient, as the spectrum of activation of these cells is complex and plastic. The authors should use additional markers and in any case, be mindful of overinterpretation based on a handful of markers.

We agree with this reviewer's comment and we have now implemented a more thorough analysis of macrophage activation, by including the analyses of expression of the functionally relevant markers Arginase1 (typically M2 associated) and iNOS (typically associated to M1). The data confirmed that CI-deficient tumors recruit more M2 macrophages in general (Fig. 7A), whereas the polarization of M1->M2 remains either comparable or more prominent in WT tumors (Supp. Fig. 16), confirming that the higher abundance of macrophages in CI-deficient tumors is not due to their

capacity to more efficiently polarize macrophages from M1 to M2 population. The text of the manuscript has been modified accordingly (line 360-373), as well as the Methods and Supplementary data explaining the gating strategies applied.

8- As the authors infer that infiltration of macrophages into the tumor bulk mediates the compensatory phenotype in NDUFS3 KO tumors allowing to acquire a more malignant phenotype, it would be important to assess the intratumoral and peritumoral localization of CAFs and macrophages at d10, and determine whether these cells infiltrate the tumors bulk to favour the acquisition of adaptive mechanisms that may participate to revert tumor indolence, or if they are present within the tumor mass in the early onset of CI-deficient neoplasia.

The contribution of fibroblasts at day 10 is not consistent among the cell lines used in this study (Supp. Fig. 11B), which is why we did not furthermore focus on their involvement in CI-deficient tumor progression. On the other hand, the Figure 6D shows F4/80 staining in tumors extracted at day 10 (now we added data for day 30 as well), meaning that macrophage recruitment is an early event in tumor progression of CI-deficient masses. However, this does not allow to conclude whether what we observe is an adaptation to lack of HIF-1 α or an intrinsic property of CI-deficient cells. It is known that hypoxia most often occurs when 2mm³ are surpassed, which means that hypoxic adaptation is required already from the very early onset. The rationale that the macrophage recruitment in CI-deficient tumors is a compensatory mechanism to overcome the lack of HIF1-mediated angiogenesis was based on the well accepted data from the literature, which associate TAMs with proangiogenic roles. We have now included additional data showing that the number of F4/80 cells is decreased in CI-deficient tumors in which HIF-1 α activity is complemented by HIF-1 α -TM (Supp. Fig. 13D), suggesting that, when cancer cell mediated HIF-1 α signaling is present, the requirement for macrophage protumorigenic activity is reduced. Moreover, we added data on Endomucin and SMA staining of clodronate or vehicle-treated CI-deficient tumors in Rag mice, which show no difference in the overall number of vessels in the tumor between the two experimental groups, but do reveal a specific reduction of SMA-negative vessels appearing in the tumor component of clodronate treated masses (Supp. Fig. 16C). Moreover, in tumors grown in nude mice, clodronate-treated CI-deficient masses were completely white (Fig. 7C), indicating lack of vascularization.

We understand that these data are not a formal proof that macrophage recruitment in CI-deficient masses is a compensatory response to the lack of HIF1 activity. Therefore, we have modified the manuscript accordingly, to acknowledge the possibility that macrophage recruitment may be the intrinsic mechanism supporting progression of CI-deficient masses, which we demonstrate to be a valid anticancer target in metformin-treated tumors.

9- The tumor growth curves presented in Fig 7E and Fig 1A are strikingly different, this is concerning- why did the authors decide to change from nude to Rag1-/- mice?

We thank the reviewer for this comment since it allows to clarify that the cytometry and histology experiments evaluating the tumor stroma were performed both on Rag and nude mice, and generally no striking difference in innate immune cell recruitment was noticed. Thus, for simplification, throughout the manuscript we showed microenvironment characterization only from the nude mice setting. However, in Rag mice, the NDUFS3 KO cells were forming larger masses than in the nude strain (most likely because of the higher immunodeficiency of the Rag strain), so we decided to use the Rag strain for the clodronate treatment, in order to be able to work with more sample material. This allowed us to observe lower number of SMA- vessels in the clodronate-treated CI-deficient masses, which supports the hypothesis that macrophages are involved in compensating the lack of HIF1 activation during CI-deficient tumor progression. We have now performed the same experiment with the HCT cell line in nude mice and confirmed that clodronate treatment reduces

progression of CI-deficient tumors. In particular, in tumors grown in nude mice, clodronate-treated CI-deficient masses were extremely small (<10mm³ after 45 days) and completely necrotic, preventing us from drawing conclusions on angiogenesis. The manuscript text, Fig. 7B and Supp. Fig. 17A-C were updated accordingly.

10- The clodronate liposome treatment is likely to affect CI-proficient tumors as well, this comparison should be included here, and also performed in the HCT model. Moreover, the depletion of monocyte/macrophages using this strategy, or others more efficient (eg CSF-1R inhibition) should be done in nude mice, and integrated when the increase in macrophage numbers is observed in CI deficient tumors (between d10 and d30).

We have now performed an experiment in which HCT CI-competent and CI-deficient cancers were both treated with clodronate and we do observe an anti-tumorigenic effect of clodronate in control tumors as well. As the reviewer suggested, this is not surprising, as it is known that TAMs accumulate in hypoxic tumor regions, where they contribute to tumor angiogenesis and invasiveness. Indeed, our control models do display hypoxia, as well as macrophage accumulation on the growing front (Fig. 6D), indicating that the macrophage function is in general crucial for tumor progression in solid tumors.

However, the effect of clodronate on CI-deficient cells is far more striking. Whereas the mice carrying clodronate-treated CI-competent masses formed frank tumors, in histology similar to their untreated controls, HCT KO xenografts treated with clodronate, even after 45 days, still formed masses not larger than 10mm³, which were completely white, indicating no blood perfusion, and on histology showed extensive necrosis, which is normally absent from CI-deficient masses (Fig. 7B). These data have now been integrated in the manuscript, and confirm that CI-deficient tumors are more sensitive to macrophage depletion.

Regarding the use of CSF1R inhibitor, with the aim to optimize the animal number and address the suggestion regarding CSF1R inhibition of the other reviewer, we have moved directly to analyze the effect of this macrophage inhibitor on metformin-treated masses, also to tackle the clinical relevance of our findings. We tested the effects of a combined therapy, by using metformin and PLX-3397 (a CSF1R inhibitor suggested by the reviewer). We confirmed that the combination of metformin with PLX-3397 was the most efficacious therapy that dramatically decreased tumor growth *in vivo*, increasing the efficacy of using a single drug. These new *in vivo* experiments have now been added to the paper in the Results section and the data are shown in Fig. 7D. The Abstract and the Discussion have now been modified accordingly.

Specific comments:

- The authors should justify the use of 143B and HCT116 cell lines for their study, is it only based on the aggressiveness of these tumor cell lines?

We believe that metabolic anti-cancer approaches that target hypoxic adaptation, such as CI inhibition, may potentially have an effect on any solid tumor. Thus, the aim in this study was to identify phenotypes developed upon CI KO, which are generalizable. This is why we chose cell lines deriving from completely different origin (mesenchymal versus epithelial). Their aggressive behavior was indeed also a criterion for their choice, as well as their ploidy for NDUFS3, as they harbor two copies of the gene and this allows a more straightforward gene editing procedure. A sentence has now been added to clarify the point.

- While in the VIKD drosophila model, authors show that CC3 is not altered, comparison of cell death is not reported in the murine models, this should be corrected to ascertain that the inhibition

on tumor growth are limited to proliferative defects and Ki67 or another proliferation read out should be performed in the drosophila model.

We have now performed the requested analyses, and would like to thank the reviewer for the suggestion. Proliferation in Drosophila tumors was assessed, by using the mitotic index marker PH3, revealing a reduced PH3 staining in CI-deficient (Igf^{-/-}Ras^{V12}V1^{KD}) tumors. The manuscript text and Supp. Fig. 6 were accordingly modified. We also now report the cell death type in murine models through an analysis of the caspase cleavage in xenografts (Supp. Fig. 3B), confirming that tumor growth is due to proliferative defects and not activation of apoptosis.

- The effect of TM-HIF mutant complementation in NDUFS3 KO cells on vasculature looks rather partial (Supp Fig 8D). Authors should perform quantitation and show additional representative images.

Images of higher quality are now shown in Supp. Fig. 10D where the vasculature rescue can be better appreciated in HIF-1 α -TM xenografts. Moreover, the quantification of the vessel number has been included in the panel and the figure legend has been modified accordingly.

- The original scans should be included in Fig 7B, and quantitation should be provided.

We have now presented the unprocessed films on which it may be appreciated that MIF and IL-8 were undetectable by the chemokine array, so the fold-change quantification is not possible. However, we did present a western blot analysis of MIF protein and qRT-PCR for MIF gene in Supp. Fig. 13, which corroborate the decrease of this factor in CI-deficient xenografts.

- Typo line 160- 'cancer' not 'caner'

This was corrected.

- Higher magnification images should be shown in Fig 3C to better observe the lack of HIF1 nuclear localization.

Higher magnification images have been provided.

- Size bars are missing in multiple representative images

We have now specified magnification in all histology panels.

- Line 430-sentence is missing a word

We here fail to understand what the reviewer is referring to.

Reviewers' Comments:

Reviewer #1:

Remarks to the Author:

Much of my questions and those raised by the other reviewer have been addressed. I would recommend though that the authors tone down any reference to "aggressive cancers" and avoid generalizing these concepts. Their mechanistic studies are impactful and largely validated by other studies in the field, but given that all of their work is in established cell line xenografts, it is hard to make broader utility conclusions.

Reviewer #2:

Remarks to the Author:

The authors have performed a significant amount of work to address all concerns raised in the previous review round, added multiple and compelling experiments which has significantly improved the paper and enriched the message and findings of this study.

REVIEWERS' COMMENTS:

Reviewer #1 (Remarks to the Author):

Much of my questions and those raised by the other reviewer have been addressed. I would recommend though that the authors tone down any reference to “aggressive cancers” and avoid generalizing these concepts. Their mechanistic studies are impactful and largely validated by other studies in the field, but given that all of their work is in established cell line xenografts, it is hard to make broader utility conclusions.

We have removed reference to aggressive cancers from the title, abstract and the main text. Moreover, in concluding remarks we acknowledged that our findings may be of importance in at least certain types of cancer, toning-down generalization of the concepts.

Reviewer #2 (Remarks to the Author):

The authors have performed a significant amount of work to address all concerns raised in the previous review round, added multiple and compelling experiments which has significantly improved the paper and enriched the message and findings of this study.